# Hsp70-associated chaperones have a critical role in buffering protein production costs

Zoltán Farkas[1†]*, Dorottya Kalapis[1†], Zoltán Bódi[1], Béla Szamecz[1], Andreea Daraba[1], Karola Almási[1], Károly Kovács[1], Gábor Boross[1], Ferenc Pál[1], Péter Horváth[1], Tamás Balassa[1], Csaba Molnár[1], Aladár Pettkó-Szandtner[2,3], Éva Klement[3], Edit Rutkai[4], Attila Szvetnik[4], Balázs Papp[1], Csaba Pál[1]*

[1]Synthetic and Systems Biology Unit, Institute of Biochemistry, Biological Research Centre of the Hungarian Academy of Sciences, Szeged, Hungary; [2]Institute of Plant Biology, Biological Research Centre of the Hungarian Academy of Sciences, Szeged, Hungary; [3]Laboratory of Proteomic Research, Biological Research Centre of the Hungarian Academy of Sciences, Szeged, Hungary; [4]Division for Biotechnology, Bay Zoltán Nonprofit Ltd, Budapest, Hungary

**Abstract** Proteins are necessary for cellular growth. Concurrently, however, protein production has high energetic demands associated with transcription and translation. Here, we propose that activity of molecular chaperones shape protein burden, that is the fitness costs associated with expression of unneeded proteins. To test this hypothesis, we performed a genome-wide genetic interaction screen in baker's yeast. Impairment of transcription, translation, and protein folding rendered cells hypersensitive to protein burden. Specifically, deletion of specific regulators of the Hsp70-associated chaperone network increased protein burden. In agreement with expectation, temperature stress, increased mistranslation and a chemical misfolding agent all substantially enhanced protein burden. Finally, unneeded protein perturbed interactions between key components of the Hsp70-Hsp90 network involved in folding of native proteins. We conclude that specific chaperones contribute to protein burden. Our work indicates that by minimizing the damaging impact of gratuitous protein overproduction, chaperones enable tolerance to massive changes in genomic expression.
DOI: https://doi.org/10.7554/eLife.29845.001

*For correspondence:
farkas.zoltan@brc.mta.hu (ZáF);
cpal@brc.hu (CPá)

†These authors contributed equally to this work

Competing interests: The authors declare that no competing interests exist.

## Introduction

Optimal allocation of cellular resources is a central concept in cell biology (*Basan et al., 2015*; *Hui et al., 2015*). Protein biosynthesis consumes a huge amount of energy: an estimated 30–50% of the energy consumption of dividing cells is dedicated to translation of the proteome (*Buttgereit and Brand, 1995*; *Russell and Cook, 1995*). Therefore, surplus protein production incurs a substantial fitness cost. As the ratio of unneeded protein reaches 30% of total protein in bacteria, ribosomes are destructed and growth is completely inhibited (*Dong et al., 1995*). Protein burdens (i.e. protein overexpression costs) are most relevant shortly after an environmental change, and are subsequently reduced once the translation has adjusted to their novel steady-state level (*Shachrai et al., 2010*). Deciphering the key molecular mechanisms that shape protein burden is an important challenge for systems biology. Moreover, this problem has biotechnological relevance as well. Protein engineering efforts towards microbial production of a single heterologous protein are often problematic, as full induction of the engineered constructs frequently yields bacteria with limited or no growth (*Kurland and Dong, 1996*).

**eLife digest** Proteins are vital for almost every process that keeps cells alive. They are made from chains of small molecules called amino acids, which need to fold into three-dimensional structures for the protein to become active. Specific molecules called chaperones help the proteins to fold properly.

However, to produce proteins a lot of energy is needed. Therefore, this process is tightly coordinated with the needs of the cells to conserve energy. If too much protein is made, it can put a burden on cells and harm the organism, even when it is a protein with no apparent cellular activities. This can be a problem under stressful conditions, for example, when cells are exposed to heat or lack nutrients.

For researchers who want to engineer cells to produce different or additional proteins, this poses a great challenge, as the modified cells often grow slowly or not at all. Until now, it was not known why proteins are harmful when produced in excess. To investigates this, Farkas, Kalapis et al. modified the cells of baker's yeast to overproduce an unneeded protein. The yeast cells were then exposed to different environmental stresses, such as too much heat or lack of nutrients, and scanned for any damage. Moreover, any potential protein burden was also measured in a collection of different cells in which each lacked one dispensable gene.

The results showed that when enough nutrients where present, producing too much of the protein only mildly affected cell growth. However, when exposed to different stressors, the cells grew more slowly. When Farkas, Kalapis et al. then blocked specific chaperones, the proteins could no longer fold properly and consequently, the cells became very sensitive to when the protein was produced in bulks.

This study shows that chaperones or environmental stress can shape protein production costs. A next step will be to investigate how sensitive other species are to protein burden, and what the underlying molecular mechanisms might be. A better understanding of how environmental and genetic factors affect the way the organisms deal with excess proteins may help to improve engineered protein-production systems in the future.

DOI: https://doi.org/10.7554/eLife.29845.002

Gene expression costs are frequently not due to the detrimental activity of unnecessary proteins, as reduced viability was observed with the overexpression of proteins with no apparent cellular activities (*Andrews and Hegeman, 1976*; *Dong et al., 1995*; *Kurland and Dong, 1996*; *Stoebel et al., 2008*; *Scott et al., 2010*). Most notably, a recent systematic study in baker's yeast (*Saccharomyces cerevisiae*) measured the copy number limit of gene overexpression across all protein coding genes (*Makanae et al., 2013*). Dosage sensitive genes were generally highly expressed, and replacement of the open reading frame of these genes with a green fluorescent protein (GFP) left the fitness cost largely unaltered. Studies in bacteria (*Stoebel et al., 2008*) and yeast (*Kafri et al., 2016*) demonstrated that growth impairment results from the process of protein production and not due to accumulating the unneeded protein product per se.

Protein production of an unneeded protein consumes nutrients and has a high energetic demand. Associated costs may arise at the level of transcription due to waste of nucleotides incorporate into RNA or occupation of RNA polymerases. Translation of the unneeded proteins may be especially costly, as it wastes amino acids, charged tRNAs and occupies ribosomes. It has been shown that these two major limiting factors of protein production vary across environments, depending on the availability of nutrients (*Kafri et al., 2016*). Transcription dominates protein burden in low phosphate, while translation dominates costs in low nitrogen conditions. Hypothetically, unneeded proteins may also overload the cellular systems involved in protein folding and degradation. Yet, the role of chaperone networks in contributing to protein burden has remained unexplored.

In this work, we show that accumulation of an unneeded protein in yeast (*S. cerevisiae*) has a relatively mild impact on fitness when nutrients are in excess and no internal or external stresses are present. However, impairment of specific molecular chaperones rendered yeast cells sensitive to gratuitous protein overproduction.

## Results

### Impact of protein burden on fitness

Recent works showed that the fitness costs associated with expressing unneeded fluorescent proteins do not result from protein toxicity or impaired metabolic processes, indicating that it is the outcome of a limitation in the protein production process itself (*Makanae et al., 2013*; *Kafri et al., 2016*). In this paper, we employ yEVenus, a rapidly folding and non-toxic YFP (yellow fluorescent protein) variant (*Sheff and Thorn, 2004*) to study protein burden. Using this protein has several advantages for our study: the amino acid composition of yEVenus and the yeast proteome are highly similar to each other (Pearson's correlation, r = 0.6477, p<0.01) and it is codon optimized specifically for yeast studies. Accordingly, toxicity of yEVenus due to misfolding is negligible. We expressed yEVenus in *S. cerevisiae* from single, low and high-copy-number plasmids, respectively, (*Gietz et al., 1988*) all of which are under the control of a strong constitutive promoter (pHSC82, see Materials and methods). The control strain carried the same vector backbone without the yEVenus open reading frame. Fitness of each genotype was determined by measuring colony size on synthetic selection medium agar plates (for further details, see Materials and methods). Cost is defined as the reduction in fitness of yEVenus overexpressing genotype relative to fitness of control cells in the same synthetic drop-out medium. When expressed from a single copy plasmid, yEVenus had no detectable fitness cost, while it caused a small, but significant 2.5% fitness decline expressed from a high-copy (2 μ) plasmid (*Figure 1A*). A denaturing polyacrylamide gel electrophoresis analysis (PAGE) indicated that when expressed from the high-copy plasmid, yEVenus constitutes ~3.7% of the total cellular proteome (*Figure 1B*).

### Genome-wide mapping of genes that mitigate protein burden

The above results indicate that accumulation of an unneeded protein in the cell has a relatively mild impact on fitness when nutrients are in excess and no internal or external stresses are present. However, such robustness to protein burden may be restricted to certain conditions: many genetic and environmental factors could potentially shape the associated fitness costs. To identify genes modulating protein burden, we performed a genetic interaction screen using the synthetic genetic array (SGA) approach (*Tong and Boone, 2006*) with the query strain carrying the yEVenus multi-copy plasmid. The screen involved construction of high-density arrays of double mutants by crossing the query mutation (yEVenus overexpression plasmid) against an array of ~5000 viable null mutants. We simultaneously measured yEVenus fluorescence intensity and fitness in all genotypes studied. Using a robotized replicating system, fitness was estimated by measuring colony size on solid agar media. Digital images were processed to calculate colony sizes, and potential systematic biases in colony growth were eliminated (see Materials and methods). Deviation of the double-mutant fitness from the product of the corresponding single-mutant fitness values was used to assess genetic interaction scores (ε, *Figure 1C*, *Supplementary file 1*). Biomass-normalized fluorescence level had no major impact on the distribution of genetic interaction scores (*Figure 1D*). This pattern was not due to any major deviation from wild type cell size (*Figure 1—figure supplement 1A*). This indicates that genetic interactions reflect a change in the fitness cost, but not in the extent of protein overexpression.

As the aim of this study was the identification of genes that mitigate the fitness costs of yEVenus overexpression, we focused on negative genetic interactions (ε < 0), i.e. when the double mutant has a lower fitness than would be expected from the product of the single-mutant fitness values. At an ε = - 0.05 cutoff value (and using a p<0.05 cutoff based on bootstrap analysis), 184 genes showed such interactions with yEVenus. By definition, lack of these genes substantially increased the fitness cost of yEVenus overexpression (*Figure 1E*). A functional enrichment analysis revealed that these genes are preferentially involved in translation, transcriptional control (e.g. transcription termination and elongation), mitochondria-related processes, and protein folding (*Table 1*, *Figure 1—figure supplement 1B*). Remarkably, deletion of genes encoding specific chaperones caused a 2–4 fold increment in the fitness costs of yEVenus overexpression (*Figure 1E*).

Enrichment of the above functional categories was not found in the set of genes showing positive genetic interactions with yEVenus overexpression. It is worth noting however a specific case, where positive genetic interaction was especially strong. Deletion of *RPI1, a* specific repressor of the Ras-

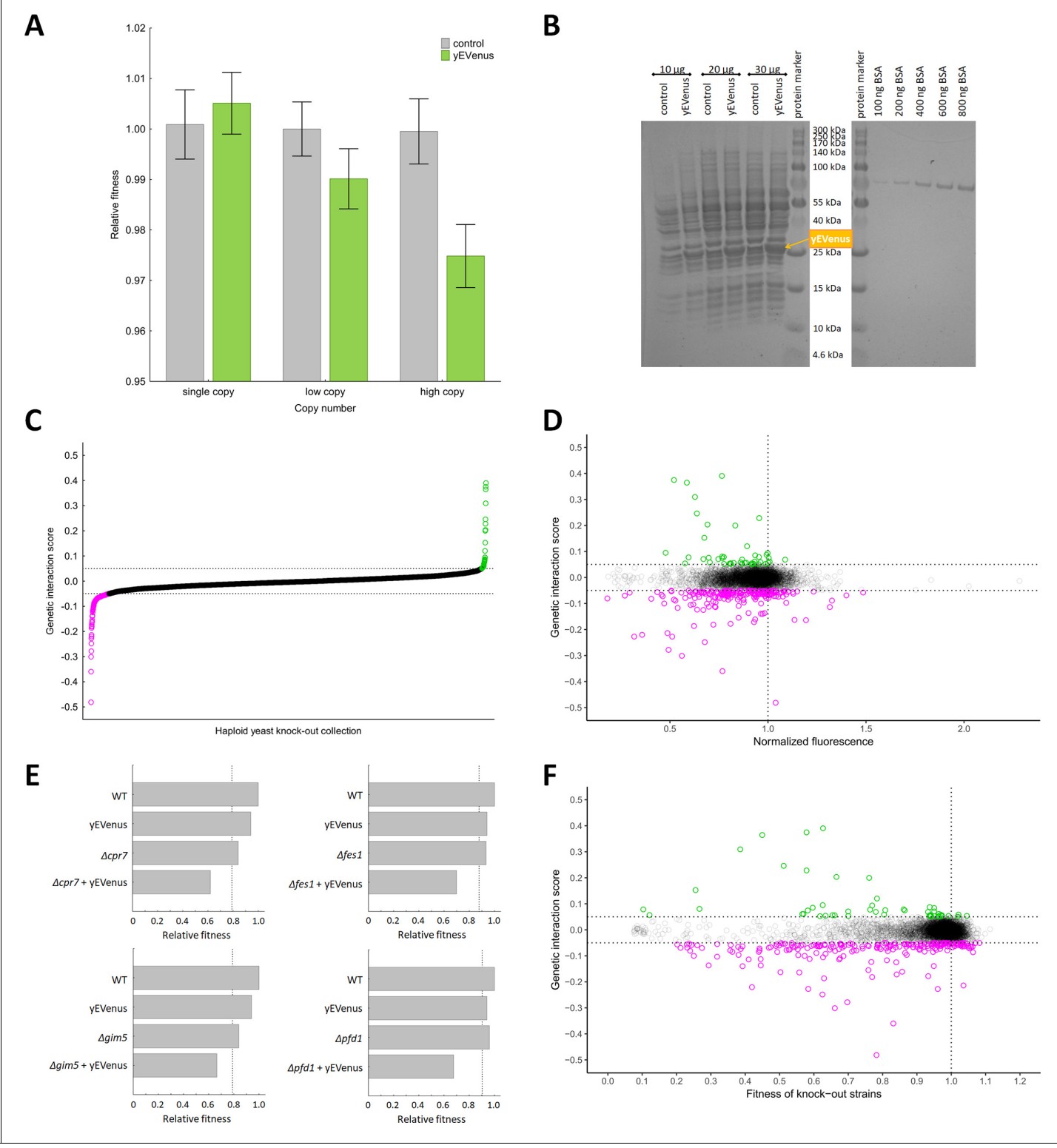

**Figure 1.** Genetic perturbation analyses. (**A**) Protein burden changes with copy number. The bar plot shows the relative fitness of yEVenus overexpressing and control genotypes as a function of plasmid copy number, a proxy of gene expression level. From a single copy plasmid, yEVenus has no detectable fitness cost (*t*-test, p=not significant), while it confers around 2.5% fitness disadvantage from a high-copy plasmid (*t*-test, p<0.001). Absolute fitness was estimated by measuring colony size after 48 hr of growth on solid medium. Relative fitness was calculated by normalizing to the absolute fitness of the genotypes with the corresponding empty vectors, respectively. The bars indicate mean ±95% confidence interval, based on at least 12 technical and 10 biological replicates each. Source file is available as *Supplementary file 5*. (**B**) PAGE analysis of whole cell protein extracts.
*Figure 1 continued on next page*

*Figure 1 continued*

The figure shows the PAGE (polyacrylamide gel electrophoresis) separation of whole cell protein extracts (10 µg, 20 µg, and 30 µg) from both the control and the yEVenus overexpressing strains in denaturing conditions (4–20% gradient Tris-Glycine SDS-PAGE). To create a standard curve, a bovine serum albumin (BSA) dilution series (100–800 ng) was loaded onto the same gel. On the basis of a densitometry analysis using the standard curve, the yEVenus (band at 27 kDa) constitutes 3.7% of the total cellular proteome when expressed from a high-copy plasmid (for further details, see Materials and methods). (C) Distribution of genetic interaction scores (ε) across the haploid yeast knock-out collection. The ε value for the vast majority of the knock-out strains is approximately zero, indicating no specific genetic interaction of the corresponding gene with yEVenus overexpression. The dashed lines on the y axis represent cutoff values for ε (0.05 and −0.05, respectively). Negative/positive interactions are color-coded as magenta/green. For the calculation of genetic interaction score, see Materials and methods. Source file is available as *Supplementary file 1*. (D) Scatterplot of the genetic interaction scores and biomass-normalized fluorescence levels of the deletion strains from the haploid yeast knock-out collection. On the x axis, one represents the wild type fluorescence level (dashed line). The dashed lines on the y axis represent the previously defined interaction value cutoffs (0.05 and −0.05, respectively). The fluorescence level of the genotypes shows only very weak correlation with the strength of the interaction (Pearson's correlation test, r = 0.05, p<0.001). Negative/positive interactions are color-coded as magenta/green. For the calculation of genetic interaction score and for the evaluation of fluorescence level, see Materials and methods. Source file is available as *Supplementary file 1*. Additional analysis of genetic interaction scores and fluorescence levels are shown in *Figure 1—figure supplement 1A and B*. (E) Examples on negative genetic interactions between single gene deletions and yEVenus overexpression. The bar plots show the relative fitness values (normalized to wild type) of single mutants (yEVenus overexpression or single gene deletions), and double mutants (deletion +yEVenus overexpression), based on six replicates. Negative deviation of the observed double mutant fitness from the expected value (designated as dashed line, calculated by the multiplicative model using the two single mutant fitness values) is referred to as negative interaction. Absolute fitness was estimated by measuring colony size after 48 hr of growth on solid medium. The deleted genes (Δcrp7, Δfes1, Δgim5, Δpfd1) are selected members of the chaperone system. Source file is available as *Supplementary file 1*. An example of positive genetic interaction is shown in *Figure 1—figure supplement 1C* (F) Scatterplot of the genetic interaction scores and the fitness of the deletion strains from the haploid yeast knock-out collection. On the x axis, one represents the wild type fitness (dashed line). The dashed lines on the y axis represent the previously defined interaction value cutoffs (0.05 and −0.05, respectively). Negative/positive interactions are color-coded as magenta/green. The fitness of the deletion strains shows only a weak positive correlation with the strength of interaction (Pearson's correlation test, r = 0.12, p<0.001). For the calculation of fitness and genetic interaction score, see Materials and methods. Source file is available as *Supplementary file 1*.

DOI: https://doi.org/10.7554/eLife.29845.003

The following figure supplement is available for figure 1:

**Figure supplement 1.** Additional analyses of the genetic interaction screen.

DOI: https://doi.org/10.7554/eLife.29845.004

cAMP pathway removed protein burden (*Supplementary file 1*, *Figure 1—figure supplement 1C*). The underlying molecular mechanisms need further investigation.

Protein synthesis is frequently limited by the availability of free ribosomes (*Vind et al., 1993*). Therefore, excess proteins occupy free ribosomes, which could be better used for the translation of native proteins. Therefore, impairment of genes involved in translation should increase protein burden. We investigated this issue further by measuring fitness in the presence of a translation inhibitor chemical agent. Cycloheximide binds the ribosome and inhibits eEF2 mediated translocation during translation (*Obrig et al., 1971*). In agreement with expectation, partial inhibition of translation elongation by cycloheximide treatment elevated protein burden (*Figure 2A*).

Similarly, inactivation of genes involved in transcriptional elongation (*HPR1, DST1, CDC73, ELP3*) significantly increased protein burden. To validate this result, we tested the effect of a transcriptional elongation inhibitor on protein burden. Mycophenolic acid interferes with nucleotide biosynthesis (*Costa and Arndt, 2000*), through inhibiting IMP dehydrogenase (IMPDH). It thereby reduces the endogenous GTP/UTP and stalls RNA polymerases. Treatment of cells with sub-inhibitory concentration of this chemical agent significantly enlarged protein burden (*Figure 2B*).

Another source of protein burden may arise due to wastes of cellular resources, including ATP and amino acids needed for protein synthesis. Indeed, inactivation of amino acid metabolism genes (*AAT2, BAT2, CYS3, PRS3, LEU3*) influenced protein burden (*Supplementary file 1*), suggesting that protein burden depends on the availability of amino acids in the environment. It was indeed so: depletion of amino acids in the growth medium increased the fitness cost (*Figure 2C*). Moreover, genes with mitochondria-related functions, including mitochondrial translation (e.g. *MRPS9*, *MRPL22*), mitochondrial DNA replication and growth (e.g. *MMM1*), mitochondrial distribution and morphology (e.g. *MDM38*) are on the gene list identified by the SGA analysis (*Supplementary file 1*).

**Table 1.** Functional enrichment analysis of genes showing synergistic interactions with yEVenus overexpression.

At an ε = - 0.05 cutoff value (and using a p<0.05 cutoff based on bootstrapping), 184 genes showed negative interactions with yEVenus. This gene set was tested for GO Slim category enrichment. A GO category was termed as enriched significantly, if the genes annotated to a particular GO term were significantly overrepresented (Fisher's exact test, odds ratio >1, p<0.05, FDR-corrected p<0.1) in the given gene set using the complete list of screened genes as background. N indicates the number of negative interacting genes belonging to the corresponding GO Slim category. Source file is available as **Supplementary file 1**.

| Go.id | Term | N | Odds ratio | P value | FDR corrected P value |
|---|---|---|---|---|---|
| GO:0002181 | cytoplasmic translation | 11 | 2.27 | 1.52E-02 | 1.69E-01 |
| GO:0006325 | chromatin organization | 15 | 1.70 | 4.65E-02 | 2.93E-01 |
| GO:0006353 | DNA-templated transcription, termination | 3 | 4.17 | 4.60E-02 | 2.93E-01 |
| GO:0006354 | DNA-templated transcription, elongation | 16 | 8.43 | 2.73E-09 | 2.73E-07 |
| GO:0006360 | transcription from RNA polymerase I promoter | 6 | 5.86 | 1.24E-03 | 3.10E-02 |
| GO:0006366 | transcription from RNA polymerase II promoter | 29 | 2.29 | 2.01E-04 | 6.71E-03 |
| GO:0006397 | mRNA processing | 9 | 2.84 | 7.77E-03 | 1.11E-01 |
| GO:0006414 | translational elongation | 5 | 3.31 | 2.45E-02 | 2.34E-01 |
| GO:0006457 | protein folding | 10 | 3.06 | 3.26E-03 | 5.43E-02 |
| GO:0007005 | mitochondrion organization | 28 | 2.59 | 4.13E-05 | 2.07E-03 |
| GO:0009408 | response to heat | 7 | 3.17 | 1.05E-02 | 1.31E-01 |
| GO:0009451 | RNA modification | 7 | 2.53 | 2.91E-02 | 2.34E-01 |
| GO:0016570 | histone modification | 8 | 2.33 | 3.05E-02 | 2.34E-01 |
| GO:0032543 | mitochondrial translation | 12 | 2.96 | 1.78E-03 | 3.55E-02 |
| GO:0048308 | organelle inheritance | 5 | 3.23 | 2.68E-02 | 2.34E-01 |

DOI: https://doi.org/10.7554/eLife.29845.005

Taken together, results of genetic and chemical perturbations of specific cellular subsystems demonstrate that impairment of transcription, translation and amino acid availability increase protein burden.

Finally, one may argue that growth rate reduction per se - irrespective of the exact nature of the environmental or genetic perturbation - may imply elevated protein burden upon overexpression. However, this is unlikely to be so, for three reasons. First, the functional roles of genes that showed genetic interactions were far from being random (*Table 1*). Second, and more generally, the correlation between the fitness of the deletion strains and the strength of the genetic interaction was very weak (*Figure 1F*). Finally, despite major differences in growth rates of yeast grown on glucose, galactose or raffinose as sole carbon sources, the relative fitness costs of protein burden remained unchanged (*Figure 2D*).

## Molecular chaperones shape protein burden

The genetic interaction screen revealed that molecular chaperones are overrepresented in the list of genes that influence protein burden. Most notably, the list includes several members of the Hsp40-70-110 complex (*FES1*, *SSE1* and *YDJ1*), and an Hsp70-90 scaffold protein (*STI1*). These Hsp70-associated proteins are functionally highly related (*Rizzolo et al., 2017*), and all play critical roles in the ATPase activation and the nucleotide exchange regulation of the Hsp70 class Ssa chaperones. Accordingly, impairment of these proteins decreases the activity of Ssa chaperones and thereby perturbs the recognition and clearance of misfolded proteins. As a consequence, aggregated proteins accumulate in the cell (*Mayer, 2013*; *Clerico et al., 2015*).

Based on these findings we hypothesized that molecular chaperones have a critical role in buffering protein burden. Several further observations support the hypothesis. First, we tested the impact of temperature stress on protein burden, not least because genes (e.g. *CPR7, YDJ1*) involved in the GO term 'response to heat' were on the list of negative genetic interactions. Subjecting yeast cells to high temperature causes a severe proteotoxic stress as it induces protein misfolding of nascent proteins and perturbs proteome homeostasis (*Trotter et al., 2002*). As expected, protein burden

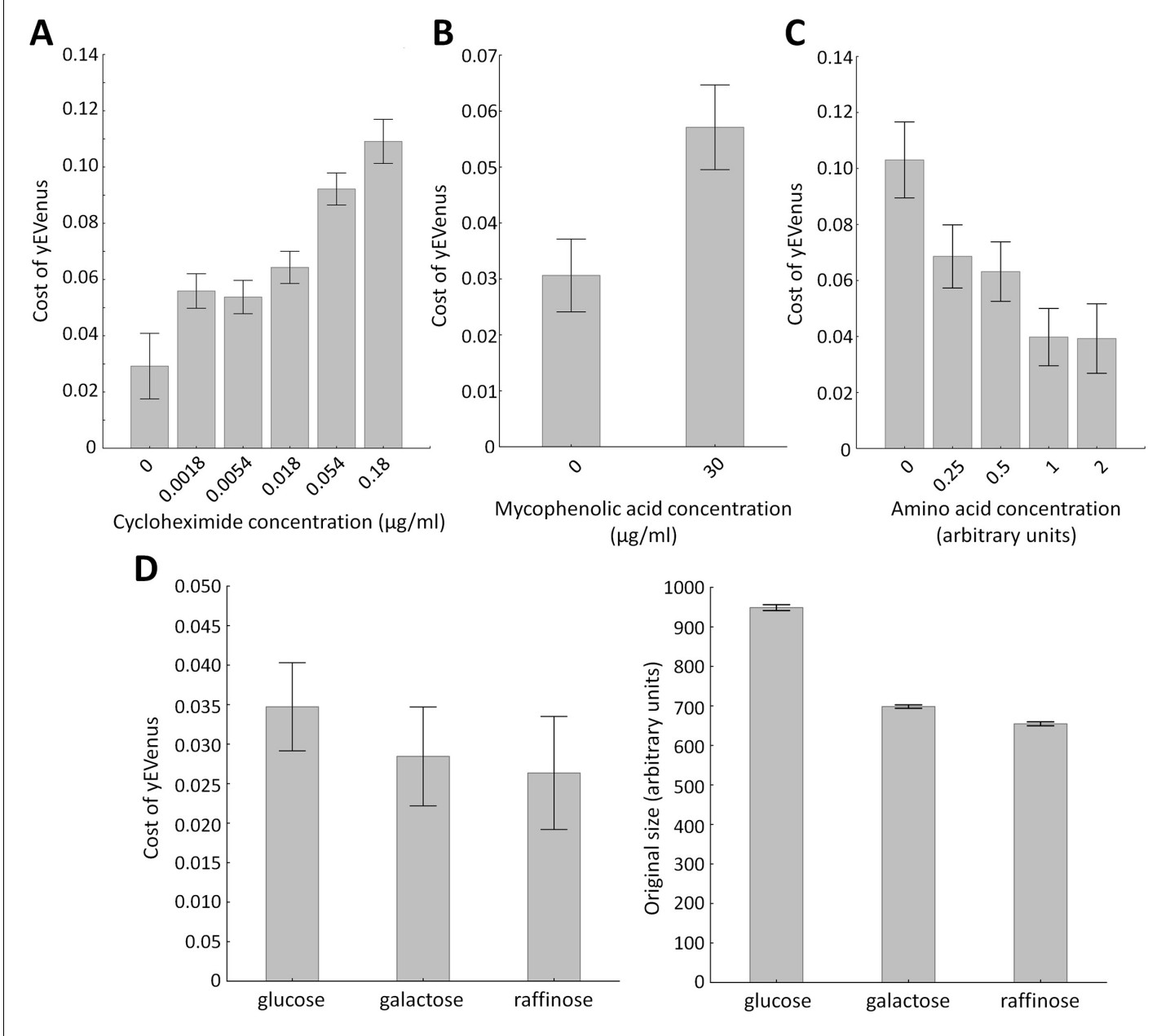

**Figure 2.** Environmental screens under protein burden. (**A**) Impact of translation inhibition on protein burden in wild type yeast. The bar plot shows the cost of yEVenus in wild type strain as a function of increasing cycloheximide concentration. Cycloheximide is a widely used chemical agent to inhibit translation. Treatment of cells with sub-inhibitory concentration (0.18 µg/ml) of this chemical agent leads to a 3.7-fold increase in protein burden (*t*-test, p<0.001). For the calculation of fitness cost of yEVenus, see Materials and methods. The bars indicate mean ±95% confidence interval, based on four technical measurements of 17 biological replicates for each concentration. Source file is available as *Supplementary file 5*. (**B**) Impact of transcription inhibition on protein burden in wild type yeast. The bar plot shows the cost of yEVenus in wild type strain in response to mycophenolic acid (MPA) stress. MPA is a well-known transcription elongation inhibitor. Treatment of cells with sub-inhibitory concentration (30 µg/ml) of this chemical agent leads to a two-fold increase in protein burden (Mann Whitney *U*-test, p<0.001). For the calculation of fitness cost of yEVenus, see Materials and methods. The bars indicate mean ±95% confidence interval, based on at least 12 technical measurements of 15 biological replicates for each concentration. Source file is available as *Supplementary file 5*. (**C**) Impact of amino acid availability on protein burden. The bar plot shows the cost of yEVenus in wild type strain as a function of amino acid concentration. Auxotrophic amino acids were supplied at normal concentration to the medium, while non-auxotrophic amino acids were serially diluted from the regular one. Arbitrary units are relative concentrations normalized to the regular amino acid level. Total depletion of non-essential amino acids (0 arbitrary unit) from the growth medium resulted in a 2.5-fold increase in protein burden, compared to the regular one (*t*-test, p<0.001). For the calculation of fitness cost of yEVenus, see Materials and methods. The bars indicate mean ±95% confidence interval, based on at least five technical measurements of 12 biological replicates for each condition. Source file is available as
*Figure 2 continued on next page*

*Figure 2 continued*

**Supplementary file 5**. (D) The impact of protein burden across different carbon sources. The left panel shows the cost of yEVenus in wild type strain on different carbon sources. The right panel shows the absolute fitness (arbitrary units estimated by measuring colony size on solid agar media) of the yEVenus overexpressing wild type strain on different carbon sources. Growth media with alternative carbon sources (respirato-fermentative galactose, respirative raffinose) led to a reduction of absolute fitness by 27–32% (right panel, *t*-test, p<0.001), compared to that on the standard carbon source (fermentative glucose). However, the relative fitness cost of yEVenus overexpression (left panel) was not affected by the change of carbon source. Specifically, the cost of yEVenus on glucose is comparable to that on galactose (*t*-test, p=0.14) or raffinose (*t*-test, p=0.07). For the calculation of absolute fitness and fitness cost of yEVenus, see Materials and methods. The bars indicate mean ±95% confidence interval, based on at least 12 technical measurements of 15 biological replicates for each of the genotype. Source file is available as **Supplementary file 5**.
DOI: https://doi.org/10.7554/eLife.29845.006

significantly increased with rising temperature (*Figure 3A*). Reassuringly, these results are insensitive to the exact promoter employed for the expressional control of yEVenus (*Figure 3—figure supplement 1A*, *Figure 3—figure supplement 1B*, *Figure 3—figure supplement 1C*, *Supplementary file 2*).

Second, as mistranslation during protein synthesis promotes misfolding and protein aggregation (*Lee et al., 2006*; *Yang et al., 2010*; *Paredes et al., 2012*), reduction of translation fidelity should also exacerbate the fitness deficit caused by protein overproduction. Reassuringly, *CTK2* and *CTK3*, two genes involved in controlling the fidelity of translation elongation (*Röther and Strässer, 2007*) were on the list of genes showing negative genetic interaction with yEVenus overexpression (*Supplementary file 1*).

Third, induction of protein misfolding by a chemical agent enhanced protein burden. We studied the cellular response to misfolded proteins generated by azetidine-2-carboxylic acid (AZC) stress (*Shichiri et al., 2001*). AZC is a toxic analog of proline, and incorporation of this chemical agent into proteins causes misfolding (*Trotter et al., 2002*; *Albanèse et al., 2006*). Application of sub-lethal dosages of AZC elevated the fitness costs associated with yEVenus overproduction (*Figure 3B*).

The fourth piece of evidence comes from monitoring cellular aggregation. An established method (*Kaganovich et al., 2008*) was utilized to measure the misfolding propensity of a fluorescently-tagged reporter protein (VHL-mCherry). Active quality-control machinery in the wild type yeast prevents misfolding of the reporter protein, leading to its uniform distribution in the cell. However, when the protein folding machinery is impaired or becomes overloaded, the reporter protein misfolds and becomes spatially sequestered. As the fluorescent tag of the reporter protein remains fully functional, protein aggregation spots within the cell become easily visible as fluorescent foci (*Kaganovich et al., 2008*).

In wild type cells, protein misfolding propensity did not increase significantly upon protein burden (*Figure 3C*). This is in line with expectation, as the fitness cost of protein overexpression in wild type was only around 2.5% (*Figure 1A*). The situation was very different when genotypes impaired in protein folding (Δfes1, Δsse1, Δsti1, Δydj1, Δpfd1, Δgim5, Δcpr7) were considered, all of which showed negative genetic interactions with yEVenus overexpression. In these genotypes, protein burden elevated the propensity of protein misfolding (*Figure 3C*, *Figure 3—figure supplement 1D*).

## Protein burden perturbs the Sti1p interaction network

The above results indicate a crucial role of the Hsp70-associated molecular chaperones in mitigating protein burden. Why should this be so? One possibility is that the unneeded proteins bind to key regulators of the Hsp70-associated chaperones which otherwise would be used to navigate folding of native proteins within the cell. To investigate the feasibility of this idea, we performed a GFP co-immunoprecipitation (co-IP) assay to identify weak in vivo physical interactions between yEVenus and native cellular proteins.

In order to extract cellular proteins without disturbing physical interactions, we used an established protocol (*Visweswaraiah et al., 2011*) specifically designed for the identification of weak protein-protein interactions. Total protein extracts from mid-exponential growth phase were immunopurified (IP) using anti-GFP antibody coupled magnetic beads and the IP-purified proteins were then subjected to LC-MS/MS analysis (see Materials and methods). Relative abundance of individual proteins in the samples was estimated by retrieving peptide counts of the individual proteins.

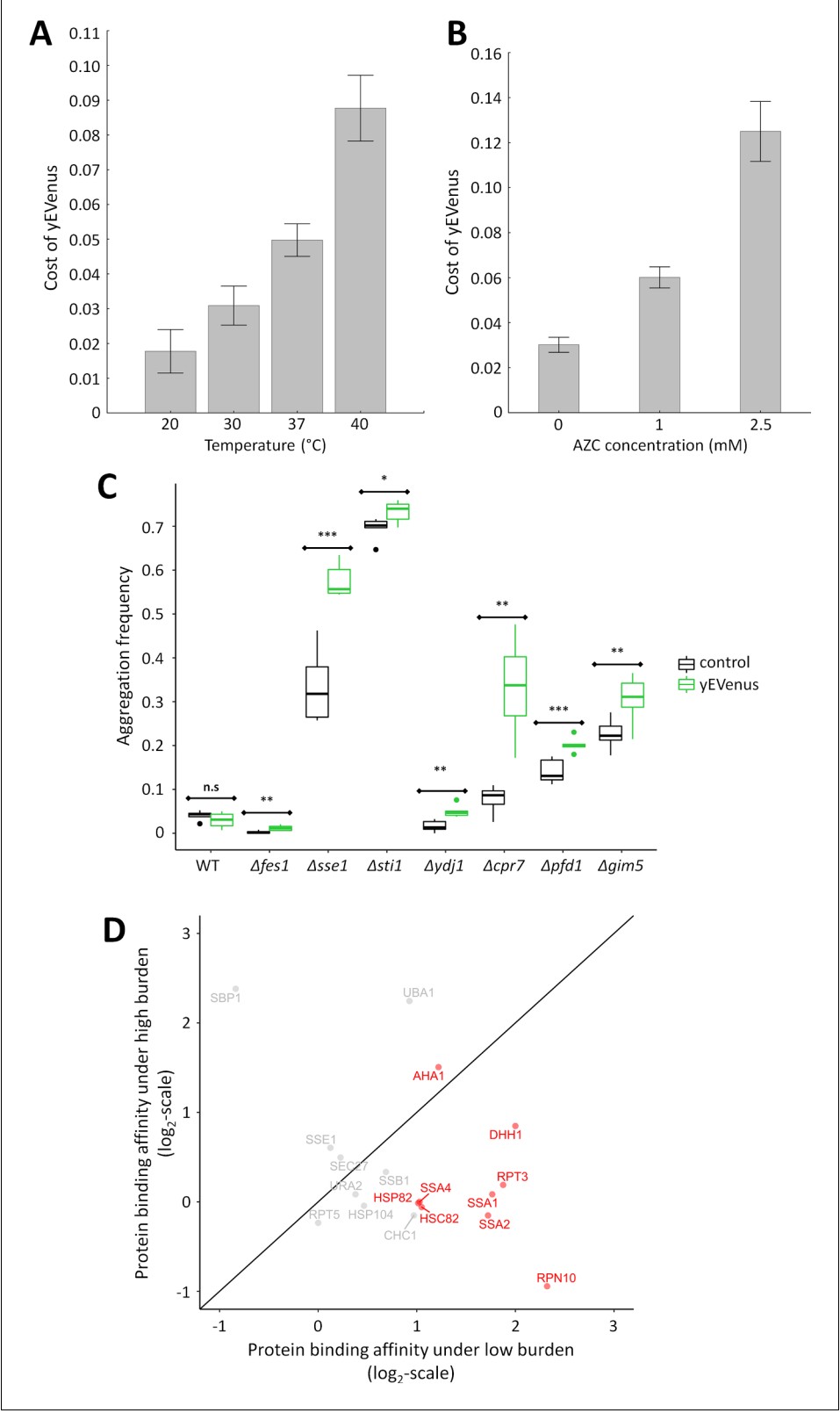

**Figure 3.** Link between protein burden and proteotoxic stress. (A) Impact of heat stress on protein burden. The bar plot shows the fitness cost of yEVenus in wild type strain as a function of increasing temperature. Protein burden significantly increased with rising temperature, resulting in a 2.8-fold difference when colonies were subjected to 40°C, in comparison to the optimal incubation temperature (30°C; $t$-test, p<0.001) The bars indicate mean ±95% confidence interval, based on at least 12 technical measurements of 15 biological replicates for each condition. Source file is available as

*Figure 3 continued on next page*

*Figure 3 continued*

*Supplementary file 5*. Additional analysis of protein burden across five different yEVenus plasmids are shown in *Figure 3—figure supplement 1A–C* (B) Impact of proteotoxic stress on protein burden. The bar plot shows the fitness cost of yEVenus in wild type strain as a function of azetidine-2-carboxylic acid (AZC) concentration. AZC is a toxic analog of proline, incorporation of this compound into newly synthesized proteins leads to misfolding in consequence of reduced protein stability. Incubation with sub-lethal dosage of AZC (2.5 mM) leads to a more than 4-fold increase in protein burden (*t*-test, p<0.001). For the calculation of fitness cost of yEVenus, see Materials and methods. The bars indicate mean ±95% confidence interval, based on at least 12 technical measurements of 15 biological replicates for each concentration. Source file is available as *Supplementary file 5*. (C) Protein aggregation propensity in yEVenus overexpressing genotypes. The bar plot shows the aggregation frequency in the wild type and in four deletion mutant strains, with (yEVenus) or without (control) protein burden. The deleted genes are selected members of the chaperone system. Protein burden by yEVenus promotes protein aggregation further in the chaperone deficient backgrounds. Aggregation frequency is 5–670% larger in the chaperone deletion mutants under protein burden, in comparison with the corresponding isogenic control strain with empty vector, respectively. The aggregation propensity in the wild type is at the same level either with or without protein burden. The frequency of cells with aggregated foci corresponds to the level of protein aggregation. Aggregation frequency was calculated as follows: the number of cells containing fluorescent foci was divided by the number of fluorescent cells in total, monitoring at least 2000 cells. For further details, see Materials and methods. The bars indicate mean ±95% confidence interval, based on at least five technical measurements for each of the genotype. Student *t*-test was used to assess difference in aggregation frequency between control and yEVenus overexpressing genotypes. */**/*** indicates p<0.05/0.01/0.001; n.s indicates p=not significant. Source file is available as *Supplementary file 5*. Representative images of VHL-mCherry localization in yeast cells are shown in *Figure 3—figure supplement 1D*. (D) Changes of Sti1p interaction partners in response to protein burden. The figure shows the scatterplot of the log(2) protein-binding affinity of 18 putative interaction partners (*Cherry et al., 2012*) of Sti1p under low- and high protein burden, respectively. Protein-binding affinity to Sti1p was estimated by calculating the peptide count fold change of Sti1p IP samples relative to the negative control IP samples both under low and high protein burden (see Materials and methods). The red points mark proteins that specifically associate with Sti1p under low protein burden. The continuous line represent x = y. Source file is available as *Supplementary file 4*.

DOI: https://doi.org/10.7554/eLife.29845.007

The following figure supplement is available for figure 3:

**Figure supplement 3.** Additional investigation of protein burden across different yEVenus plasmids and different chaperone-deficient mutants.
DOI: https://doi.org/10.7554/eLife.29845.008

After applying several filtering steps (see Materials and methods), we identified 34 proteins that bind to yEVenus (*Supplementary file 3*). Altogether, the list of putative interacting partners includes five proteins with chaperone-related functions (*Supplementary file 3*). Notably, Sti1p and Ydj1p not only binds to yEVenus, but were identified also in the genetic interaction assay. Both proteins are involved in the activation of Ssa proteins, key components of the Hsp70 complex.

The above results indicate that as a globular protein, yEVenus binds weakly, but significantly to certain molecular chaperones and to Sti1p in particular (*Supplementary file 3*). This raises the possibility that the protein burden is linked to perturbation of the native physical interactions of Sti1p by yEVenus. To investigate this issue, we performed a reciprocal co-IP assay with the aim to identify quantitative changes in physical interactions of Sti1p in response to protein burden. Accordingly, we used a strain that expresses a C-terminally epitope-tagged Sti1p (Sti1p-3xFLAG) and investigated it both under low and high protein burden. Total protein extracts from mid-exponential growth phase were immunopurified (IP) using anti-FLAG antibody coupled beads and the IP-purified proteins were then subjected to LC-MS/MS analysis, as previously (Materials and methods).

The analysis focused on 18 proteins, all of which have been described to physically interact with Sti1p in prior studies (*Cherry et al., 2012*). Our method confirmed half of these 18 protein interactions under low protein burden, that is when yEVenus was expressed from a single-copy plasmid (*Figure 3D*, *Supplementary file 4*). Remarkably, we observed a significant drop in protein-binding affinity of Sti1p with as many as 8 out of the nine detected interaction partners under high protein burden (*Figure 3D*, *Supplementary file 4*). Most notably, protein-binding affinity decreased by 70%, 75% and 55% in the cases of Ssa1, Ssa2p and Hsp90p, respectively. This is all the more significant, as these proteins are exceptionally important and well-characterized interaction partners of Sti1p (*Chen and Smith, 1998*; *Song and Masison, 2005*; *Balchin et al., 2016*). Finally, protein-binding between yEVenus and Sti1p was detectable under high protein burden only (*Supplementary file 4*). We speculate that promiscuous binding of Sti1p with certain globular proteins (such as yEVenus) has no functional consequences unless the cellular dosage of the partner protein exceeds a critical threshold. Collectively, these data suggest that protein burden promotes a partial disassociation of interaction partners from Sti1p, putatively leading to partial disassociation of the Hsp70-Hsp90 chaperone complex.

## Discussion

Our work demonstrates that even gross accumulation of an unneeded gratuitous protein in the cell has a relatively mild impact on fitness when no internal or external stresses are present (*Figure 1A*). However, such robustness to protein burden was restricted to specific conditions only. We explored the molecular mechanisms underlying robustness to protein overproduction. Our main findings are as follows.

First, deletion of genes involved in translation, transcriptional control, and mitochondria-related processes rendered yeast cells hypersensitive to protein overexpression. Our observation that translational and transcriptional perturbations modulate protein burden was validated further by chemical and environmental stress screens, and is also consistent with prior studies (*Kafri et al., 2016*). Therefore, protein burden varied substantially across genetic backgrounds and environmental stresses. We note that mutants with impaired mitochondria exhibit reduced respiratory growth, and therefore they have to rely on less efficient modes of ATP production. However, beyond ATP production, mitochondria are involved in the synthesis of certain amino acids as well (*Ahn and Metallo, 2015*; *Zong et al., 2016*). Therefore, future works should elucidate the exact molecular mechanisms underlying the elevated protein burden in cells deficient in mitochondrial functions.

Second, prior studies suggested that expression of an unneeded protein effectively decreases the fraction of proteome allocable to ribosomes and useful biosynthetic proteins, thereby causing a growth defect (*Scott et al., 2010*). In principle, mutations could therefore modulate protein burden by simply increasing the proteome fraction of the unneeded protein. However, the fractional contribution of yEVenus to the total proteome was not elevated in gene knock-out strains (*Figure 1D*, *Figure 1—figure supplement 1A*). This indicates that allocation models that rely on transcription and translation only cannot fully account for protein burden.

Third, and most significantly, an interacting chaperone network shapes protein burden (*Figure 4*). The Hsp70 complex is a key player in the maintenance of normal proteostasis. The soluble Ssa proteins (members of the Hsp70 family) recognize and associate transiently with exposed hydrophobic patches of misfolded proteins in the cytosol and prevent protein aggregation (*Mayer, 2013*; *Clerico et al., 2015*). Deletion of specific activators (*YDJ1*, *STI1*, *FES1* or *SSE1*) of Ssa proteins substantially elevated protein burden, and resulted in protein aggregation. Indeed, Ssa protein's capacity to bind and release client proteins heavily depends on these activators (*Wegele et al., 2003*). In particular, the nucleotide exchange factors (Sse1p and Fes1p) are responsible for client-release and thereby support the refolding or the proteasomal degradation of misfolded proteins (*Gowda et al., 2013*). It is worth noting that due to partial functional redundancy of Ssa proteins (*Hasin et al., 2014*), the corresponding *SSA* genes did not emerge in the screen. In agreement with expectation, temperature stress, elevated mistranslation rate and a chemical misfolding agent all substantially enhanced protein burden. We conclude that molecular chaperones have an important role in buffering protein burden.

Finally, we found evidence that yEVenus - a typical, globular fluorescent protein binds to Sti1p, one of the key regulators of the Hsp70-Hsp90 complex (*Song and Masison, 2005*; *Wolfe et al., 2013*). We hypothesize that Sti1p may be especially prone to promiscuous protein binding, as it has an over 2-fold higher fraction of unstructured residues than the proteome average (data not shown). Approximately, half of Sti1p putative physical interacting partners (*Cherry et al., 2012*) are involved in the maintenance of normal proteostasis. The list includes members of the Hsp70-Hsp90 complex, Hsp104 disaggregase, proteasome subunits and ubiquitin-associated proteins. Therefore, one might expect that perturbation of Sti1p interactions by a highly abundant, weakly interacting protein (*Figure 3D*) would have serious fitness consequences in times of proteotoxic stress. Future works should elucidate this hypothesis further and specifically the role of promiscuous peptide binding in protein burden.

Our work has important implications for future studies. The distribution of genomic expression generally follows a highly skewed power-law like distribution with a small number of exceptionally highly expressed genes (*Ueda et al., 2004*; *Lu and King, 2009*). Highly expressed genes contain various cost-minimizing gene architectures (*Frumkin et al., 2017*). Such genes are under especially severe selective constraints, possibly to avoid misfolding and consequent formation of protein aggregates (*Geiler-Samerotte et al., 2011*). Even though highly expressed proteins are not particularly prone to misfolding, they may still indirectly influence protein aggregation in the cell.

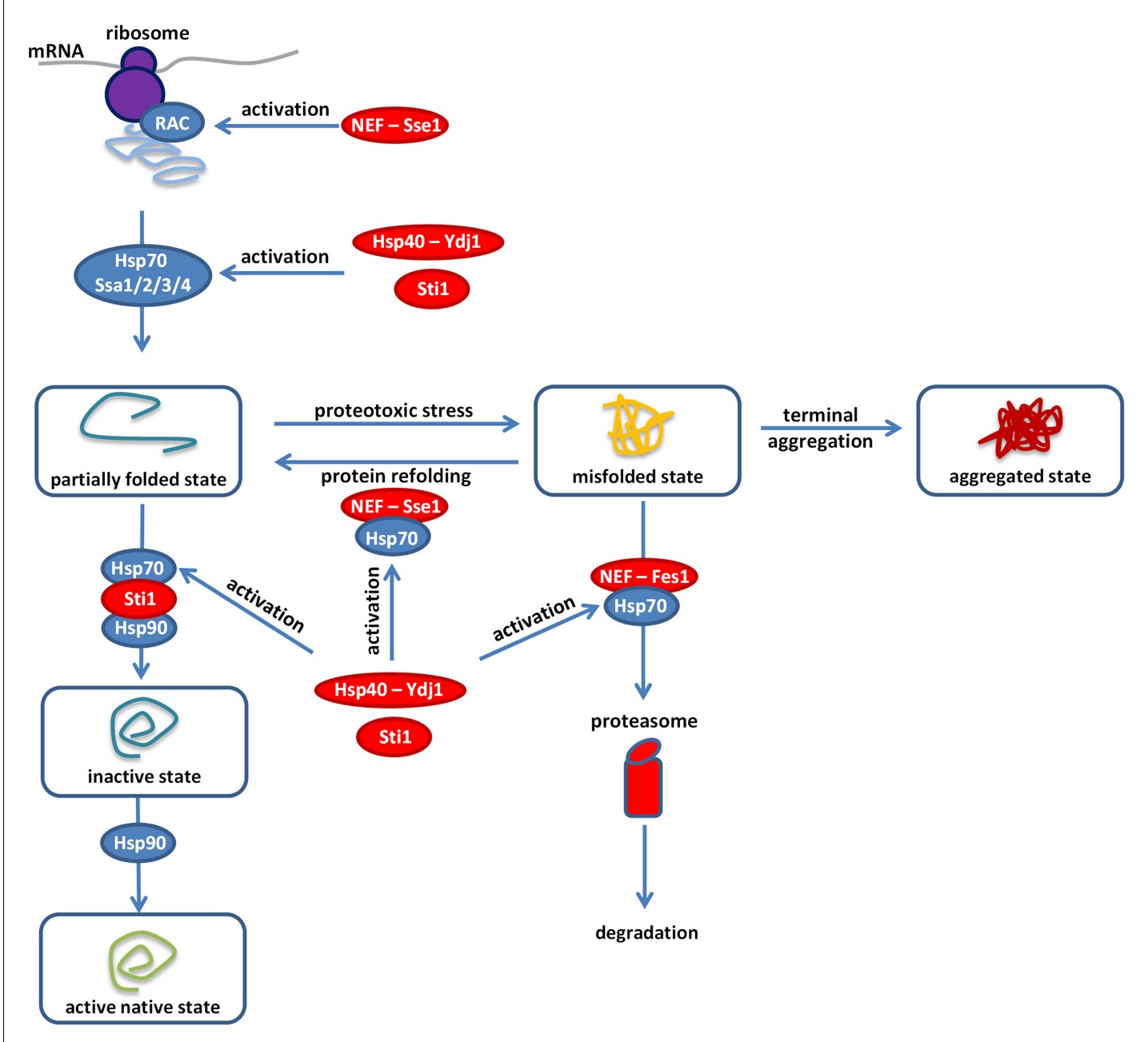

**Figure 4.** Overview of the proteostasis network. Malfunction of the protein quality control system impairs the proteome balance by driving cellular proteins into toxic metastable (partially folded or misfolded) conformations from their correctly folded native state (*Balchin et al., 2016*). Accumulation of these folding intermediates could further overload this surveillance system and could lead to the collapse of the proteostasis network. Hypothetically, overexpression of a gratuitous protein (such as the yEVenus) might not be tolerated in a misfolding sensitized background, as it could add an extra-layer of threat to the cell. Our genome-wide genetic interaction screen (SGA) revealed the importance of a central regulatory complex to buffer overexpression costs. This complex maintains the normal activity of the Ssa chaperones (members of the Hsp70 family) that act on misfolded proteins. In addition, one member of this complex also acts on the ribosome-associated complex (RAC). Inactivation of the constituent members (Hsp70-90 scaffold Sti1p, Hsp40-Ydj1p, NEF-Sse1p, and NEF-Fes1p, color-coded as red) of this complex exacerbated the cost of yEVenus overexpression. In such genetic backgrounds, the clearance of misfolded proteins by protein refolding or proteasomal degradation is affected. In agreement with the genetic perturbation screen, conditional induction of proteotoxic stress in the yEVenus overexpressing wild type strain also exaggerated the cost of the overexpression. Remarkably, based on physical interaction assays, we found evidences that protein burden perturbs the interaction network of Sti1p, putatively leading to a dysfunctional Hsp70-Hsp90 chaperone complex. As a consequence, downregulation of the proteostasis network is expected, which would have serious fitness consequences in times of proteotoxic stress.

DOI: https://doi.org/10.7554/eLife.29845.009

Specifically, our work raises the possibility that highly expressed proteins bind to key components of the chaperone network which otherwise would be used to navigate folding of other native proteins within the cell. As a consequence, the availability of active chaperone molecules decreases, leading to increased propensity for damaging protein aggregation, especially in times of proteotoxic stress. It is important to emphasize that yEVenus is a codon optimized fluorescent protein (*Sheff and Thorn, 2004*), and is not particularly prone to misfolding and consequent toxicity (*Kafri et al., 2016*). Therefore, this hypothesis is conceptually distinct and complementary to the issue of whether aggregation-prone proteins impose a fitness cost through toxicity (*Plata et al., 2010*; *Geiler-Samerotte et al., 2011*).

More generally, several molecular chaperones can buffer the damaging effects of protein mutations (*Csermely, 2001*; *Queitsch et al., 2002*; *Cowen and Lindquist, 2005*; *Paaby and Rockman, 2014*). Chaperone overload by highly expressed proteins may influence this process. In a similar vein, it appears that protein burden depends on genetic variation and environmental conditions as well. Therefore, the cellular capacity to tolerate major fluctuations in genomic expression heavily depends on the genetic makeup: the associated fitness costs should vary extensively across microbial species occupying different environmental niches. Finally, we anticipate that our genome-wide approach uncovering the determinants of protein burden will help the design of improved host strains for the efficient overproduction of recombinant proteins.

# Materials and methods

**Key resources table**

| Reagent type (species) or resource | Designation | Source or reference | Identifiers | Additional information |
|---|---|---|---|---|
| Strain (Saccharomyces cerevisiae) | Y7092 | PMID: 16118434 | | SGA query strain, mat alpha |
| Other | non-essential gene deletion collection (BY4741, MATa) | PMID:12140549 | YSC1053 | Open BioSystem (Dharmacon) |
| Other | synthetic genetic array (SGA) technique | PMID: 16118434 | | |
| Software, algorithm | ImageJ software | PMID: 22930834 | RRID:SCR_003070 | |
| Software, algorithm | Gene Onthology term enrichment with topGO (version 2.28) | PMID: 16606683 | | |
| Software, algorithm | org.Sc.sgd.db (version 3.3.0) packages in R | *Core Team, 2017* | | http://www.R-project.org |
| Software, algorithm | Machine learning-based phenotypic analysis | PMID: 21807964 | | |
| Software, algorithm | Advanced Cell Classifier | PMID: 28647475 | | http://www.cellclassifier.org/ |
| Software, algorithm | Proteome Discoverer (v 1.4) | Thermo Fisher Scientific (Germany) | | |
| Recombinant DNA reagent | pKT0090 plasmid | PMID: 15197731 | Addgene:Plasmid #8714 | contains yEVenus |
| Recombinant DNA reagent | YEplac181 plasmid | PMID: 3073106 | Addgene:Plasmid #8628 | high copy plasmid |
| Recombinant DNA reagent | YCplac111 plasmid | PMID: 3073106 | Addgene:Plasmid #53249 | single copy plasmid |
| Recombinant DNA reagent | pRS315 plasmid | PMID: 2659436 | ATCC 77144 | low copy plasmid |
| Recombinant DNA reagent | pFA6a-TEV-6xGly-3xFlag-HphMX plasmid | Tim Formosa | Addgene:Plasmid #44083 | |
| Recombinant DNA reagent | pGAL-VHL-mCherry | PMID: 18756251 | | galactose inducible VHL-mCherry |

*Continued on next page*

Continued

| Reagent type (species) or resource | Designation | Source or reference | Identifiers | Additional information |
|---|---|---|---|---|
| Commercial assay or kit | μMACS GFP Isolation Kit | Miltenyi Biotec (Germany) | 130-091-125 | |
| Commercial assay or kit | μMACS DYKDDDDK Isolation Kit | Miltenyi Biotec (Germany) | 130-101-591 | *DYKDDDDK is also known as FLAG tag |

## Yeast strains and plasmids

All strains used in this study were derived from the Y7092 *Saccharomyces cerevisiae* parental strain (SGA query strain: MAT alpha; *can1delta::STE2pr-Sp_his5, lyp1delta, his3delta1 leu2delta0, ura3-delta0, met15delta0*). The fluorescent yEVenus protein was transformed into the parental Y7092 strain on a high copy number plasmid (YEplac181, [*Gietz et al., 1988*]) by a standard protocol (*Gietz and Schiestl, 2007*). The transformants were selected on leucine dropout synthetic complete medium (SC-MSG, 1 g/l monosodium glutamate (Sigma-Aldrich, Germany), 1.7 g/l Yeast Nitrogen Base (BD, Germany), supplemented by amino-acid mix without leucine).

## Plasmid construction

To measure the fitness cost of protein overexpression, yEVenus, a non-toxic protein with no enzymatic activity and optimized codon usage was selected (*Sheff and Thorn, 2004*). The corresponding gene was integrated into a high copy expression vector. Heterologous promoters frequently perturb the transcription of other genes, by binding/titrating essential transcription factors, causing a skewed distribution of transcription factors. To minimize this problem, expression of yEVenus was driven by the native promoter of Hsc82p. Hsc82p is one of the most abundant cellular proteins in yeast (*Borkovich et al., 1989*; *Ghaemmaghami et al., 2003*). In contrast to many other chaperones (such as Hsp82p), it is expressed constitutively and shows only minor variation across stress conditions.

The high copy hc-Venus plasmid was constructed in three steps. First, the genomic *HSC82* gene of the *Saccharomyces cerevisiae* strain BY4741 including its promoter sequence was amplified from genomic DNA using restriction site containing oligonucleotides (*B_HSC_promoter, B_HSC82_terminator*). The product was cut with *BamH*I and *Pst*I endonucleases, and was ligated to *BamH*I and *Pst*I digested YEplac181 (*Gietz et al., 1988*) plasmid, generating the hc HSC82 construct. The promoter region was also PCR amplified with *B_HSC_promoter* primer and *HSC-promoter-HSP-orf-reverse* primer, which product was *BamH*I digested and ligated into a *BamH*I and *Stu*I digested hc_HSC82 plasmid. The resulting plasmid (pHSC_promoter plasmid) was designed to facilitate the insertion of virtually any ORF using its *Nhe*I and *Pst*I restriction sites. The *yEVenus* ORF along with the *ADH1* terminator was amplified from the pKT0090 plasmid (*Sheff and Thorn, 2004*) using *Nhe*I-Venus_ATG and *Adh1_term_primer_pst1* oligonucleotides. The given PCR product was *Nhe*I and *Pst*I digested and ligated to the identically digested pHSC_promoter plasmid. The generated plasmid (hc_Venus) was used to express yEVenus in *S. cerevisae*, under the control of the strong constitutive *HSC82* promoter. For the selection of the plasmid, *LEU2* marker was used in a leucine dropout synthetic medium. The control strains carry the original backbone plasmid (YEplac181) without the fluorescent protein.

To investigate the effect of plasmid copy number variation on protein burden, was inserted both into the *BamH*I-*Pst*I digested single (YCplac111, [*Gietz et al., 1988*]) and low copy plasmid (pRS315, [*Sikorski and Hieter, 1989*]).

Finally, to ensure that the key results are insensitive to the exact promoter used for controlling the expression of yEVenus, we constructed four extra isogenic plasmids with different, naturally occurring promoters in the yeast genome. These promoters drive the expression of cytosolic proteins (Gpp1p, Tal1p, Pdc1p, and Tdh3p), all which are as highly abundant as the constitutively expressed Hsp90p (*HSC82*, source: PeptideAtlas 2013 dataset [*Wang et al., 2012*]). Specifically, the pHSC82 region was eliminated from the hc_Venus plasmid after *Sac*I-*Nhe*I digestion. Next, the promoter regions of *GPP1, TAL1, PDC1,* and *TDH3* were amplified from wild type genomic DNA using restriction site-containing oligonucleotides (frw_*Sac*I, rev_*Nhe*I). Finally, the PCR products (pGPP1, pTAL1, pPDC1, and pTDH3) were inserted into the *Sac*I-*Nhe*I digested hc_Venus plasmid backbone.

Fluorescence level showed only minor variation across the five high copy plasmid constructs (*Figure 3—figure supplement 1A*).

## Cellular quantification of yEVenus protein

To quantify the yEVenus protein within the proteome, whole cell extracts were prepared from wild type cells, in the presence and absence of the yEVenus plasmid. Single colonies were inoculated into leucine dropout SC-MSG liquid medium, and were grown until saturation at 30°C. The saturated cultures were diluted and grown to mid-exponential phase (OD$_{600}$ = 0.8), and $10^8$–$10^9$ cells were used to extract total protein using established protocol (*Visweswaraiah et al., 2011*). Whole cell extract (WCE) concentration was determined by using Bicinchoninic Acid Kit (Sigma-Aldrich), according to the manufacturer's instructions. Whole cell extracts from the control and overexpression strain were separated on a 4–20% gradient Tris-Glycine gel (Lonza, Germany) under denaturing (SDS, sodium dodecyl sulfate) conditions, along with a dilution series (100–800 ng) of a standard protein (1 mg/ml bovine serum albumin, BSA, Sigma-Aldrich). Densitometry analysis of the protein bands on SDS-polyacrylamide gel was conducted by ImageJ software (*Schneider et al., 2012*). A standard curve was established by plotting the pixel numbers of BSA dilution series bands versus BSA concentrations. The yEVenus band (27 kDa) intensity was corrected by subtracting the intensity of the equal-sized protein band in the control strain. Based on the standard curve, the pixel number of the yEVenus band (27 kDa) was converted into concentration, and the ratio of the quantified yEVenus protein to the loaded whole cell extract was calculated.

## Synthetic genetic array analysis

To identify genes mediating yEVenus burden, we performed a synthetic genetic array (SGA) screen (*Tong and Boone, 2006*). The query mutation (in our case the yEVenus carrying plasmid) was crossed to an ordered array of ~5000 viable, non-essential gene deletion mutants (MATa; YKO collection, Open BioSystem, Dharmacon Inc, Lafayette, Colorado, United States, [*Giaever et al., 2002*]). The method applies a series of replica pinning steps onto solid medium in an automated manner, using the following series of steps: (a) selection for MATa/α diploids (SC-MSG medium (1 g/l monosodium glutamate, 1.7 g/l Yeast Nitrogen Base, supplemented by amino-acid mix) with G418 (200 μg/ml, Sigma-Aldrich) was used), (b) induced sporulation by reducing carbon and nitrogen levels in the nutrient, (c) selection for MATα meiotic progeny (*can1Δ::MFA1pr-HIS3, lyp1Δ*) using canavanine (50 mg/L, Sigma-Aldrich) and thialysine (S-(2-Aminoethyl)-L-cysteine hydrochloride, 50 mg/L, Sigma-Aldrich) containing medium, (d) selection for the query mutation (leucine dropout medium), and finally selection for the gene deletions (G418 containing medium; *KanMX4* cassette confers resistance against G418). Finally, the array of meiotic progeny harboring both mutations (yEVenus plasmid and gene deletion) was scored for fitness (see below). To evaluate genetic interactions, an array of 'single' mutants was also constructed, where the query strain harbors the control high copy plasmid (YEplac181), without the fluorescent protein ORF.

The *HIS3* (YOR202W) deletion strain (*his3::KanMX4*) was used as wild type control, for the following reasons: (1) fitness of this strain is indistinguishable from the BY4741 parental wild type strain (*Qian et al., 2012*); (2) it possesses the same selection marker (required for the SGA method) as all other single gene deletion strains; (3) it carries the *KanMX4* cassette in the nonfunctional *his3Δ1* allele.

## Quantitative fitness measurements

We developed a robust high-throughput and precise workflow for fitness measurements based on colony size. Solid media were prepared using 2% agar (2% was previously found to be optimal for reproducible colony size measurement, data not shown). The ordered arrays of strains at 384-density were replicated onto solid medium with a robotized replicating system. The system consists of a Microlab Starlet liquid-handling workstation (Hamilton Bonaduz AG, Switzerland), equipped with a 384-pin replicating-tool (S&P Robotics Inc, Toronto, Ontario, Canada) and a custom-made sterilization station for the replicating-tool. After 48 hr of acclimatization to the medium at 30°C, plates were replicated again onto the same medium and photographed after 48 hr of incubation at 30°C. Digital images were processed to calculate colony sizes. We took special care to control for potential systematic biases in colony growth, such as uneven media composition, changes in physical

parameters of incubation, or competition for nutrients between neighboring colonies (*Szamecz et al., 2014*). Colonies located next to the edges/corners of the plates and colonies with low circularity (*i.e.* circularity <0.8) were removed from further analysis. Genotype fitness was estimated by the mean fitness of six replicate colonies. The replicate number used is comparable to (eight replicates for Kuzmin et al, in preparation) or even higher than the number of replicates other studies (four replicates for ([*Hoke et al., 2008*; *Baryshnikova et al., 2010*; *Costanzo et al., 2010*]) used to estimate fitness based on colony size.

Genetic interactions score was calculated as $\varepsilon = f_{ab} - (f_a \times f_b)$, where $f_a$ and $f_b$ are quantitative fitness measures of the two single (deletion or yEVenus overexpression) mutants, while $f_{ab}$ is the fitness of the double mutant (deletion and yEVenus overexpression). Negative ($\varepsilon < 0$) and positive ($\varepsilon > 0$) interaction scores indicate that the fitness defect of the double mutant is higher and lower than expected by the multiplicative model, respectively. We applied the confidence threshold of $|\varepsilon| > 0.05$ and $p < 0.05$ to define significantly interacting gene pairs. *p*-values were calculated using the bootstrap method (*Efron and Tibshirani, 1994*), resampling $f_a$, $f_b$, and $f_{ab}$ separately. We tested the null hypothesis that $\varepsilon = 0$.

## Functional enrichment analysis

Based on the systematic genetic-genetic interaction screen, the list of genes showing negative interaction with the yEVenus overexpression (i.e. their deletions increased the fitness effect of overexpression) were retrieved and tested for Gene Ontology term enrichment with topGO (version 2.28) (*Alexa et al., 2006*) and org.Sc.sgd.db (version 3.3.0, [*Carlson, 2016*]) packages in R programming environment (*Core Team, 2017*). To focus on the important GO terms, we restricted our search to the GOSlim categories maintained by the SGD project (*Cherry et al., 2012*). A GO category was termed as enriched significantly, if the genes annotated to a particular GO term were significantly overrepresented (Fisher's exact test, odds ratio >1, $p < 0.05$, FDR-corrected $p < 0.1$) in the given gene set using the complete list of screened genes as background.

## Fitness estimates under environmental stress

Genotype fitness was estimated under control (no-stress) and different stress environments, as above. Unless otherwise indicated, all conditions used leucine dropout SC-MSG medium. The following non-lethal stress conditions were used: translation inhibition (0.0018–0.18 μg/ml cycloheximide, AppliChem GmbH, Germany), transcription inhibition (0.30 μg/ml mycophenolic acid (MPA), Santa Cruz Biotechnology, Germany), heat stress (37°C and 40°C), proteotoxic stress (1–2.5 mM azetidine-2-carboxylic acid (AZC), Santa Cruz Biotechnology), amino acid limitation (auxotrophic amino acids were supplied at normal concentration to the medium, while the non-auxotrophic amino acids were serially diluted (*i.e.* 0x - 2x of the regular concentration)). Fitness cost of yEVenus protein overproduction (proxy for protein burden) is defined by $1 - W_V/W_C$, where $W_V$ and $W_C$ indicate absolute fitness values (*i.e.* colony sizes) of the genotypes with yEVenus and control plasmids, respectively.

## Evaluation of fluorescence level across genotypes

The fluorescence level of the final SGA array strains was evaluated by measuring yEVenus signal in liquid medium. Briefly, the array of colonies were inoculated into liquid leucine dropout SC-MSG medium, and kinetic runs were initiated in a Synergy 2 fluorescence plate reader (Biotek, Winooski, Vermont, United States) for 48 hr, using the following filters: 500/27 (excitation), 528/20 (emission). During the kinetic run, the absorbance ($OD_{600}$) and yEVenus fluorescence ($\lambda_{ex}515$ nm / $\lambda_{em}528$ nm) of the growing cultures were monitored simultaneously, with time points taken every 1.5 min. For each time points, the $OD_{600}$ normalized yEVenus fluorescence (FLOD) was calculated. The fluorescence of a given strain was assessed by calculating the median of the five highest FLOD values.

## Quantitative aggregation assay

In order to quantitatively measure and compare the level of protein aggregation in the double mutants to the corresponding single mutants (i.e. deletion), an established method (*Kaganovich et al., 2008*) was applied. This method examines the condition of the protein quality-

control machinery of the cell, based on the aggregation of a fluorescently tagged (mCherry, $\lambda_{ex}$587nm/$\lambda_{em}$610nm) human protein (von-Hippel-Lindau, VHL). This human protein is prone to misfolding in the absence of its cofactor (elongin BC), which is not present in *S. cerevisiae*. Fully functional quality-control machinery can stop aggregation of VHL-mCherry, leading to disperse cytosolic localization of the fluorescence. On the other hand, an overload of the control machinery promotes VHL protein aggregation, while leaving the fluorescent tag functional. In this case, the red fluorescence appears as a puncta inside the cell, due to the sequestration of aggregated proteins into dedicated compartments. All mutants carrying the plasmid ($p_{GAL}$-VHL-mCherry-Ura) were grown until saturation in leucine and uracil dropout SC-MSG medium, containing 2% raffinose as carbon source. To induce VHL-mCherry production, the saturated cultures were diluted into leucine and uracil dropout SC-MSG medium, containing 1% raffinose and 2% galactose. After 14 hr of induction, cell fluorescence was detected by high content microscopy, using the following filter sets: excitation: 560–580 nm, emission: 590–640 nm. Images were acquired by employing an Operetta high-content screening microscope (PerkinElmer, Waltham, Massachusetts, United States). Samples were grown and images were acquired in black optical 96-well plates (Greiner Bio-One, Austria) using a 60x high-numerical aperture objective. Five image stacks were made in each well, each of which consists of 7 z-stacks ranging from −1.5 µm to 1.5 µm relative to the focal plane with 0.5 µm step size. The following custom developed image and data analysis pipeline was used. First, an image filter was applied to amplify spots and project a z-stack. Images were corrected for illumination inhomogeneities (*Smith et al., 2015*), single cells were segmented and 118 cellular features were measured based on morphology, shape and intensities. Machine learning-based phenotypic analysis was performed (*Horvath et al., 2011*; *Piccinini et al., 2017*) using supervised learning and the ratio of phenotypic classes was determined. The ratio of cells containing aggregation loci was calculated using at least 2000 cells.

## Identification of protein–protein interactions

To reveal the in vivo physically interacting protein partners of yEVenus, whole cell extracts were prepared from wild type cells in both the presence and absence of the yEVenus overexpression plasmid, and then a GFP co-immunoprecipitation (GFP co-IP) assay was performed. First, single colonies were inoculated into leucine-dropout SC-MSG liquid medium, and were grown until saturation at 30°C. The saturated cultures were diluted and grown to mid-exponential phase ($OD_{600}$ = 0.8), and $10^8$–$10^9$ cells were collected, flash frozen and used to extract total protein using an established protocol (*Visweswaraiah et al., 2011*). Protein concentration of the whole cell extract (WCE) was determined by using Bradford Protein Assay (Bio-Rad, Hercules, California, USA), according to the manufacturer's instructions. Total protein extracts (2 mg) were immunopurified (IP) using 40 µl anti-GFP antibody-coupled 50 nm superparamagnetic beads (µMACS GFP Isolation Kit, Miltenyi Biotec, Germany). The unbound material was removed by washing the beads with 2 ml (equal to 50x beads volume) detergent-free buffers as follows: three times with 1x TBS and once with 25 mM ABC ($NH_4HCO_3$) buffer. The immunopurified proteins were desalted (*Hubner et al., 2010*) after on-bead-digestion with trypsin (Promega, Germany). The LC-MS/MS analysis was performed by using a nanoflow RP-HPLC on-line coupled to a linear ion trap-Orbitrap (Orbitrap-Elite, Thermo Fisher Scientific, Germany) mass spectrometer as in a previous study (*Kobayashi et al., 2015*) with the following modification: the 20 most abundant, multiply charged ions were selected from each MS survey for MS/MS analysis.

Raw data were converted into peak lists using Proteome Discoverer (v 1.4, Thermo Fisher Scientific). First, we performed a search against the Swissprot and Uniprot databases (*Pundir et al., 2017*), taking into consideration of the sequence of yEVenus. Search parameters and acceptance criteria were set as previously published (*Kobayashi et al., 2015*). Close homologues were only reported if at least three unique peptides matched to the protein.

Spectral counting was used to estimate relative abundance of individual proteins in the samples: peptide counts of the individual proteins were normalized to the total number of peptide identifications in each sample (*Horvath et al., 2017*). Proteins (i) with reproducible detection (|$\log_2$fold-change| < 0.67 between biological replicates), (ii) with at least two identified peptides, (iii) with at least 5% coverage and (iv) with a median-normalized protein binding affinity score above a previously defined cutoff value (2 according to [*Li et al., 2016*]) were considered as proteins that specifically associate with yEVenus. Protein-binding affinity to yEVenus was estimated by calculating the

peptide count fold change of yEVenus IP (wild type strain with yEVenus plasmid) samples relative to the negative control IP samples (wild type strain with control plasmid).

Reciprocal co-immunoprecipitation (co-IP) was performed in order to investigate physical interaction partners of Sti1p. First, a PCR-based C-terminal epitope-tagging of Sti1p was performed using established protocols (*Funakoshi and Hochstrasser, 2009*). Briefly, the transformation cassette was amplified from the pFA6a-TEV-6xGly-3xFlag-HphMX plasmid (a gift from Tim Formosa, Addgene plasmid # 44083) with primers containing homology to the C-terminal of *STI1*. Transformants were selected on YPD containing 300 µg/ml hygromycin (Santa Cruz Biotechnology). Correct clones were verified by colony-PCR and subsequent capillary sequencing of the C-terminal of *STI1*. Next, the single copy (low protein burden) or high copy (high protein burden) yEVenus plasmid was transformed into the Sti1p-FLAG-tagged strain. Finally, the yEVenus expressing strains were subjected to co-IP assay.

Whole cell extraction (WCE), immunoprecipitation (IP) and washing steps were performed as above, with the following modification: to reduce the effect of protein burden, a more stringent washing step was applied using the manufacturer's (µMACS DYKDDDDK Isolation Kit, Miltenyi Biotec) 'Wash 1' buffer (150 mM NaCl, 1% Igepal CA-630, 0.5% sodium deoxycholate, 0.1% SDS, 50 mM Tris-HCl, pH 8.0). The LC-MS/MS and raw data analysis were the same as above. Close homologues were only reported if at least three unique peptides matched to the protein. The effect of protein burden on Sti1p interacting partners was investigated by comparing the protein binding affinity of these partners under low and high protein burden. Binding affinity scores below the cutoff value indicate weaker, non-specific associations of proteins with Sti1p. Protein-binding affinity to Sti1p was estimated by calculating the peptide count fold change of Sti1p IP (IP with specific antibody to FLAG) samples relative to the negative control IP (IP without specific antibody (protein A)) samples both under low and high protein burden. Proteins i) with at least two identified peptides; ii) with at least 5% coverage and iii) with a median-normalized protein-binding affinity score above a previously defined cutoff value (two according to [*Li et al., 2016*]) were considered as proteins that specifically associate with Sti1p under low protein burden.

# Additional information

### Funding

| Funder | Grant reference number | Author |
| --- | --- | --- |
| Magyar Tudományos Akadémia | Postdoctoral Fellowship Programme PD-038/2015 | Zoltán Farkas |
| Magyar Tudományos Akadémia | Postdoctoral Fellowship Programme Postdoc2014-85 | Károly Kovács |
| Gazdaságfejlesztési és Innovációs Operatív Program | GINOP-2.3.2-15-2016-00026 | Péter Horváth Balázs Papp |
| Gazdaságfejlesztési és Innovációs Operatív Program | GINOP-2.3.2-15-2016-00006 | Péter Horváth |
| Gazdaságfejlesztési és Innovációs Operatív Program | GINOP-2.3.2-15-2016-00032 | Aladár Pettkó-Szandtner |
| Magyar Tudományos Akadémia | Bolyai Research Scholarship | Aladár Pettkó-Szandtner |
| Gazdaságfejlesztési és Innovációs Operatív Program | GINOP-2.3.2-15-2016-00001 | Éva Klement |
| Seventh Framework Programme | Initial Training Network METAFLUX 264780 | Balázs Papp |
| Wellcome | WT 098016/Z/11/Z | Balázs Papp |
| Magyar Tudományos Akadémia | Lendület Programme LP 2009-013/2012 | Balázs Papp |
| Wellcome | WT 084314/Z/07/Z | Csaba Pal |
| European Research Council | 648364 H2020-ERC-2014-CoG | Csaba Pal |

| Magyar Tudományos Akadé-mia | Lendület Programme LP 2012-32/2016 | Csaba Pal |
| --- | --- | --- |
| Gazdaságfejlesztési és Innová-ciós Operatív Program | GINOP-2.3.2-15-2016-00014 | Csaba Pal |
| Gazdaságfejlesztési és Innová-ciós Operatív Program | GINOP-2.3.2-15-2016-00020 | Csaba Pal |

The funders had no role in study design, data collection and interpretation, or the decision to submit the work for publication.

## Author contributions
Zoltán Farkas, Conceptualization, Formal analysis, Validation, Investigation, Visualization, Methodology, Writing—original draft, Writing—review and editing, interpretation of data; Dorottya Kalapis, Investigation, Methodology, Writing—review and editing, interpretation of data; Zoltán Bódi, Investigation, Methodology, interpretation of data; Béla Szamecz, Conceptualization, Investigation, Methodology, interpretation of data; Andreea Daraba, Karola Almási, Investigation, Methodology, Project administration; Károly Kovács, Gábor Boross, Ferenc Pál, Software, Formal analysis, interpretation of data; Péter Horváth, Resources, Software, Formal analysis; Tamás Balassa, Csaba Molnár, Software, Formal analysis; Aladár Pettkó-Szandtner, Éva Klement, Formal analysis, Methodology, interpretation of data; Edit Rutkai, Attila Szvetnik, Resources, Methodology; Balázs Papp, Conceptualization, Formal analysis, Supervision, Writing—review and editing, interpretation of data; Csaba Pál, Conceptualization, Resources, Formal analysis, Supervision, Funding acquisition, Writing—original draft, Writing—review and editing, interpretation of data

## Author ORCIDs
Zoltán Farkas http://orcid.org/0000-0002-5085-3306
Csaba Pál http://orcid.org/0000-0002-5187-9903

## Decision letter and Author response
Decision letter https://doi.org/10.7554/eLife.29845.023
Author response https://doi.org/10.7554/eLife.29845.024

# Additional files

## Supplementary files
• Supplementary file 1. Results of the Synthetic Genetic Array with yEVenus. This file contains information of the Synthetic Genetic Array with yEVenus in single gene deletion backgrounds and was used to create the following figures/tables: *Figure 1C, D, E and F*, *Figure 1—figure supplement 1 A–C*, and *Table 1*.
DOI: https://doi.org/10.7554/eLife.29845.010

• Supplementary file 2. Genetic interaction screen using alternative promoters. The table contains fitness and genetic interaction data of chaperone-deficient mutants overexpressing yEVenus driven by alternative promoters. This file contains the dataset used to create *Figure 3—figure supplement 1C*.
DOI: https://doi.org/10.7554/eLife.29845.011

• Supplementary file 3. Results of the GFP co-immunoprecipitation assay with yEVenus. This file contains information of the GFP co-IP assay performed on the yEVenus overexpressing strain.
DOI: https://doi.org/10.7554/eLife.29845.012

• Supplementary file 4. Results of the Sti1p co-immunoprecipitation assay. This file contains information of the Sti1p co-IP assay performed on wild type and Sti1-3xFLAG strains under high and low protein burden, that is with high and low copy yEVenus plasmid, respectively. This file contains the dataset used to create *Figure 3D*.
DOI: https://doi.org/10.7554/eLife.29845.013

• Supplementary file 5. All data used to create figures in the manuscript. The file contains the datasets used to create specific manuscript figures: it contains the data for *Figures 1A*, *2A, B, C, D*, *3A, B and C*, *Figure 3—figure supplement 1A,B*.
DOI: https://doi.org/10.7554/eLife.29845.014

• Transparent reporting form
DOI: https://doi.org/10.7554/eLife.29845.015

### Major datasets

The following previously published datasets were used:

| Author(s) | Year | Dataset title | Dataset URL | Database, license, and accessibility information |
|---|---|---|---|---|
| Wang M, Weiss M, Simonovic M, Haertinger G, Schrimpf SP, Hengartner MO, von Mering C | 2013 | S.cerevisiae - Whole organism, SC (PeptideAtlas,March,2013) | https://pax-db.org/dataset/4932/242/ | Publicly available at PaxDb: Protein Abundance Database |
| Wang M, Weiss M, Simonovic M, Haertinger G, Schrimpf SP, Hengartner MO, von Mering C | 2014 | S.cerevisiae - Whole organism, SC (GPM,Aug,2014) | https://pax-db.org/dataset/4932/243/ | Publicly available at PaxDb: Protein Abundance Database |
| Pundir S, Martin MJ, O'Donovan C | 2017 | Saccharomyces cerevisiae (Baker's yeast) [4932] | https://www.uniprot.org/uniprot/?query=taxonomy:4932 | Publicly available on UniProt (https://www.uniprot.org/) |

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
