## [Decision Letter]

[Editors’ note: a previous version of this study was rejected after peer review, but the authors submitted for reconsideration. The first decision letter after peer review is shown below.]

Thank you for choosing to send your work, "Translation linked chaperones have a critical role in buffering protein production costs", for consideration at *eLife*. Your submission has been assessed by a Senior Editor and two reviewers. Although the work is of interest, we regret to inform you that the findings at this stage are too preliminary for further consideration at *eLife*.

Specifically, concerns were raised by both reviewers about the strength of evidence supporting the main conclusions as well as the extent to which the interpretation of the work is consistent with the current state of the chaperone field.

Reviewer #1:

The manuscript of Farkas et al. investigates the cellular subsystems that buffer the effects of overproduction of a non-native fluorescent protein using synthetic genetic array (SGA) methodology in *Saccharomyces cerevisiae*. Under stress-free conditions the effects of protein over-production remain minimal, however under diverse stress conditions the effect becomes enhanced. The authors show that genetic or chemical impairment of processes such as transcription, translation, amino acid biosynthesis, and protein folding causes cells to be more sensitive to protein overproduction. While the notion that over-production of unneeded proteins causes an overload to the chaperone network is intriguing, additional evidence is required before the manuscript is ready for publication (see below).

Major concerns:

1) Through SGA analysis the authors demonstrate in Figure 3 that impairment of translation and transcription enhances the toxic effects of protein overproduction. The novelty of this finding is reduced by a paper published this year, which reported translation and transcription dominant fitness costs in cells overexpressing a non-native fluorescent protein under standard laboratory growth conditions (Kafri et al., 2016.).

2) The novel and "most significant" finding of the paper that "impairment of a chaperone network involved in co-translational folding has a critical role in defining protein production costs" needs additional lines of experimental data to support their claim. (Findings summarized in the Discussion section). The data provided that elevated temperature and protein denaturing agents increases protein burden is in agreement with their hypothesis but is not conclusive. The authors hypothesize that yEVenus occupies CLIPS chaperones which otherwise would be used to fold other native proteins within the cell. This interaction could easily be investigated using co-immunoprecipitation assays to show a physical interaction between chaperone and the GFP variant. Alternatively, the authors could investigate physical interaction between CLIPS chaperones and client proteins in the absence and presence of yEVenus overexpression to show that yEVenus disrupts the physical interaction of endogenous proteins.

3) The Results section describes quantification of yEVenus to levels of 3.7% of the total cellular proteome by performing PAGE. An image of this gel should be included in the manuscript as part of Figure 2 as opposed to data not shown.

4) Subsection “Genome-wide mapping of genes that buffer protein overproduction costs” and Figure 3. The authors describe they observed differences in growth rates of yeast grown on different carbon sources without observing differences in fitness costs due to protein burden. Figure 3 plots the Cost of protein burden in the different growth conditions, however the control data should be included to show differences in growth rate. It is not sufficient to merely describe the differences in the Figure legend.

Reviewer #2:

The manuscript by Farkas et al. relates to an important question of the costs of protein synthesis. Design of experiments and the way they were conducted appear correct. Data are of high quality. Unfortunately, I see two basic problems with interpretation (points A and C). I cannot recommend this paper for publication in its present shape.

A) Table 1. Are these p-values corrected for multiplicity of comparisons? I do not see any remark on that. How many were there MIPS classes used in the analysis? If they were many then p-value of 3.83E-03 (chaperones) may be well in the hell of insignificance. Why MIPS, FunSpec and year 2002? GOrilla allows to analyse a ranked single list of genes which appears preferable over setting an arbitrary threshold to select a target group (http://cbl-gorilla.cs.technion.ac.il/). Not only chaperones but also other categories have rather unimpressive p-values. A large data set often yields results at similar levels of uncorrected significance but with hardly any meaning.

B) Translation, transcription, metabolism of RNA and mitochondria show up as enriched in nearly every study of growth effects caused by deletions under nearly any conditions. I know that here the idea is to extract interaction out of joint effect of two factors. But, I would be careful. What categories were enriched among deletions with positive epistatic effect? Have I missed these results?

C) The following statement is meant to justify why the paper is focused on co-translational chaperones: "The list (of gene deletions interacting with GFP overproduction; rev add) includes members of the Gim/prefoldin complex (GIM4/GIM5/PFD1), two Hsp90 co-chaperones (CPR7, STI1), a type I HSP40 co-chaperone involved in regulation of HSP90 and HSP70 functions (YDJ1), and a member of the Hsp70-Hsp40 chaperone complex (SSE1). The majority of these proteins are parts of a chaperone network involved in co-translational folding of thousands of client proteins. These so-called CLIPS chaperones (Albanèse et al., 2006) associate the translation apparatus with nascent polypeptides emerging from the ribosome and help de novo folding of newly synthesized proteins. Inactivation of these genes sensitizes the proteome to protein misfolding and downstream aggregation (Albanèse et al., 2006; Willmund et al., 2013)."

My objections:

1) CLIPS is not a real, functionally defined, class of proteins. It is just an operational term covering genes involved in chaperoning/folding functions which are down-regulated in their expression at the time of stress (sometimes transiently) as opposed to HSP, which are up-regulated, and was used just in one (?) publication (Albanese, 2006). The CLIPS proteins are only indirectly linked to participation in chaperoning of nascent proteins: translation is halted (transiently) when stress comes, so CLIPS may be involved in chaperoning fresh polypeptides. (Besides, STI1 is in HSP, SSE1 and YDJ1 are in between, CPR7 and FPR4 work on proline twisting together with chaperones but are not chaperones themselves.)

2) In fact, none of the listed proteins is working co-translationally. Prefoldin sounds nice but it just assists selected new proteins on their way to TRiC after they are released from a ribosome. SSE1 is somewhat linked to nascent proteins, it is a nucleotide exchange factor for SSB (co-translational Hsp70). But it works also for SSA (stress-induced Hsp70). Besides, SSE1 releases, not binds, clients of SSB, possibly after they leave ribosomes (Willmund).

3) The co-translational binding of SSB to growing polypeptides is probably facilitated by ZUO1 and SSZ1, it is a real pity these two most relevant genes are not among those spotted in this work (SSB1 and 2 had to be deleted both, one deletion does not have any effect).

4) Although SSBs are believed (rather than truly confirmed) to chaperone co-translationally, their action is poorly recognized and probably not universal. Willmund says: "Thus, co-translationally acting Hsp70 meets the challenge of folding the eukaryotic proteome by stabilizing its longer, more slowly translated, and aggregation-prone nascent polypeptides". Authors of this manuscript say that their GFP is codon-unbiased, and it is just 230 aa long, so it does not appear a suitable client for SSBs anyway. Sure, overproduced GFP can misfold and aggregate more often than when rare but unfolding of such stuff would most probably involve SSAs and many other, non-co-translational chaperones. Where are they?

5) Conclusion. The authors selected one, decade old, paper to build the co-translational interpretation. Neither subsequent work of Frydman's lab nor views broadly accepted among the chaperone folk (Hartl, 2016) support their claims. The selected set of chaperones could have appeared not because they themselves are important but rather some of their clients (perhaps listed in the same table) malfunction when they are absent. One never knows with such promiscuous molecules. Again, the statistical signal may be too weak, so there is not much to think about (see point A).

[Editors’ note: what now follows is the decision letter after the authors submitted for further consideration.]

Thank you for submitting your article "Chaperone overload increases protein production costs" for consideration by *eLife*. Your article has been reviewed by three peer reviewers, and the evaluation has been overseen by Patricia Wittkopp as the Senior Editor and Reviewing Editor. The following individual involved in review of your submission has agreed to reveal his identity: Tobias Warnecke (Reviewer #3).

The reviewers have discussed the reviews with one another and the Reviewing Editor has drafted this decision to help you prepare a revised submission.

Summary:

The manuscript by Farkas et al. investigates the cellular subsystems that buffer the effects of overproduction of a non-native fluorescent protein using synthetic genetic array (SGA) methodology in *Saccharomyces cerevisiae*. Under stress-free conditions the effects of protein over-production remain minimal, however under diverse stress conditions the effect becomes enhanced. The authors show that genetic or chemical impairment of processes such as transcription, translation, amino acid biosynthesis, and protein folding causes cells to be more sensitive to protein overproduction. They then go on to propose that molecular chaperones have a critical role in buffering protein production costs as deletion of some genes involved in molecular chaperone function increase the fitness cost associated with protein over-production. The reviewers and I are enthusiastic about this work being potentially suitable for publication in *eLife*, but also agree that some revisions are needed prior to publications. These are described below. The experiments indicated as required were felt to be essential by all three reviewers and myself for supporting the conclusions presented.

Essential revisions:

Most significant modification to the text:

1) Examine and discuss the complete set of results from the unbiased genome-wide approaches (the genetic interaction screen and the co-IP screen), not just the set of results that can be rationalized. In our opinion, these 2 approaches provide most of the novelty of this work, and so their complete results should be more prominent in the paper. For example, the nature of positive genetic interactions with YFP should also be examined for completeness. Even if no significant enrichment could be detected among the genes involved in positive interactions, the authors should still report the observation. While the authors' focus on chaperones is understandable, as it constitutes an additional cost that can be easily rationalized as part of the protein's production history, I felt distinctly short-changed by the absence of further study (or even discussion!) of other genes that revealed negative epistatic interactions. This includes genes related to mitochondrial organization/translation, explicitly listed as enriched in Table 1, but also other genes found that do not fall into significantly enriched GOSlim categories and remain entirely unmentioned.

Two additional experiments:

2) Express yEVenus from a different promoter. (Rationale: The explanation that the overexpression of YFP perturbs chaperone network by "sequestering" parts of it (as Figure 4 suggests) is appealing. However, the overexpression of YFP in the study is driven by the promoter of HSC82, a chaperone. Although the authors claim that it is relatively constitutively expressed, querying SPELL for expression profile shows that HSC82 is co-expressed with many other protein folding proteins, suggesting that it shares transcription factors with them. Adding multiple additional copies of the HSC82 on a multi-copy plasmid could therefore decrease expression of many chaperones simply by competition, making cells more sensitive to deletion of particular chaperones. Therefore, it is important to confirm at least the most important results of the paper by overexpressing YFP using a different promoter that is unrelated to chaperones.)

3) Reciprocal yEVenus-Sti1 Co-IP. (Rationale: Given the major novel finding of the manuscript relies on the chaperone network having a critical role in shaping protein production costs, it would be nice to show in the main text a validation of the physical interaction between yEVenus and Sti1 through reciprocal co-immunoprecipitation analysis. Further, their Co-IP mass spectrometry data identifies several other cellular proteins that physically interact with yEVenus and do not have a role in chaperone function, and this analysis does not identify any other chaperone related genes, which based on their conclusions in surprising. The authors make no comment of these inconsistent observations in the text.

Additional points raised by one or more reviewers that should be addressed:

4) The authors identify 184 gene deletions that show confident negative genetic interactions with overexpression of YFP. The authors report that this list of genes is enriched for ribosome biogenesis, transcriptional control, amino acid metabolism, mitochondria-related processes and protein folding (Table 1). However, none of the GO terms listed in Table 1 seem to be associated with ribosome biogenesis or amino acid metabolism. Could the authors clarify where that conclusion originated? Also, the enrichment for mitochondria-related processes is never discussed in the manuscript.

5) One of the main motivations for the study was provided by the genome-wide genetic interaction screen. The goal of this screen was to uncover gene deletions that mitigate or exacerbate the fitness cost of unneeded protein overproduction. A key assumption is that the protein load (i.e., the number of YFP molecules per cell) is approximately the same in all ~5000 deletion mutants and thus any potential fitness defects are due to the inability of the deletion mutant to cope with the load. The justification for this assumption is provided in Figure 1 where the authors plot the correlation between the genetic interaction score (i.e., the synergistic fitness gain or defect resulting from combining a deletion with YFP overexpression) as a function of "biomass-normalized fluorescence level" (i.e., YFP abundance). Given the lack of correlation, the authors conclude that "genetic interactions reflect a change in the fitness cost, but not in the extent of protein overexpression".

I'm not sure that Figure 1 addresses this question completely because it is not clear what "biomass-normalized fluorescence level" actually represents. Biomass normalization is done by dividing the total fluorescence of a yeast culture by its OD. However, OD depends both on cell number and cell size/volume, which vary across deletion mutants and may affect the total culture fluorescence in a complex way. The authors should therefore discuss this relationship in more detail and make sure that normalized fluorescence does indeed represent the extent of protein overexpression. This is especially important considered that, according to Figure 1, normalized fluorescence varies extensively across mutants. What is the most likely cause for that variation? For example, a number of mutants are known to harbor more copies of high-copy plasmids than others (e.g., bik1, bim1) – do those mutants tend to show lower normalized fluorescence?

6) The identification of Sti1 as a protein that not only influences YFP production but also physically binds YFP is interesting. However, it is important to verify how often Sti1 co-purifies with other proteins, i.e. the number of physical interactions currently reported for Sti1 in the databases.

---

## [Author Response]

[Editors’ note: the author responses to the first round of peer review follow.]

Reviewer #1:[…] Major concerns:1) Through SGA analysis the authors demonstrate in Figure 3 that impairment of translation and transcription enhances the toxic effects of protein overproduction. The novelty of this finding is reduced by a paper published this year, which reported translation and transcription dominant fitness costs in cells overexpressing a non-native fluorescent protein under standard laboratory growth conditions (Kafri et al., 2016.).

Please note that we explicitly refer/ed to the corresponding work both in the previous and in the new version of the manuscript:

“Our observation that perturbations to translation and transcription modulate protein burden is consistent with a recent report showing that both processes are limiting in yeast grown under standard media (SC) conditions (Kafri et al., 2016).”

“Our observation that translational and transcriptional perturbations modulate protein burden was validated further by chemical and environmental stress screens, and is also consistent with prior studies (Kafri et al., 2016).”

2) The novel and "most significant" finding of the paper that "impairment of a chaperone network involved in co-translational folding has a critical role in defining protein production costs" needs additional lines of experimental data to support their claim. (Findings summarized in the Discussion section). The data provided that elevated temperature and protein denaturing agents increases protein burden is in agreement with their hypothesis but is not conclusive. The authors hypothesize that yEVenus occupies CLIPS chaperones which otherwise would be used to fold other native proteins within the cell. This interaction could easily be investigated using co-immunoprecipitation assays to show a physical interaction between chaperone and the GFP variant. Alternatively, the authors could investigate physical interaction between CLIPS chaperones and client proteins in the absence and presence of yEVenus overexpression to show that yEVenus disrupts the physical interaction of endogenous proteins.

Thank you for the suggestion. Indeed, such experimental evidence for our claims regarding to the chaperone overload was absent in the previous version of this manuscript. Therefore, to support our hypothesis, we performed a GFP co-immunoprecipitation assay and found direct evidence for a physical interaction between the yEVenus and a key chaperone regulator. We write: “Finally, we found evidence that yEVenus, a typical, globular fluorescent protein, physically interacts with Sti1p, one of the key regulators of SSA proteins.” […]“As Sti1p has multiple roles in the activation of the Hsp70 complex, one might expect that a drop in the availability of free Sti1p would have serious fitness consequences in times of proteotoxic stress.”

3) The Results section describes quantification of yEVenus to levels of 3.7% of the total cellular proteome by performing PAGE. An image of this gel should be included in the manuscript as part of Figure 2 as opposed to data not shown.

Done. The image of the PAGE gel is now included as Figure 1.

4) Subsection “Genome-wide mapping of genes that buffer protein overproduction costs” and Figure 3. The authors describe they observed differences in growth rates of yeast grown on different carbon sources without observing differences in fitness costs due to protein burden. Figure 3 plots the Cost of protein burden in the different growth conditions, however the control data should be included to show differences in growth rate. It is not sufficient to merely describe the differences in the Figure legend.

Done. The figure showing the absolute fitness (arbitrary units calculated from colony size) differences on different carbon sources is now included as a sub-panel of Figure 3.

Reviewer #2:The manuscript by Farkas et al. relates to an important question of the costs of protein synthesis. Design of experiments and the way they were conducted appear correct. Data are of high quality. Unfortunately, I see two basic problems with interpretation (points A and C). I cannot recommend this paper for publication in its present shape.A) Table 1. Are these p-values corrected for multiplicity of comparisons? I do not see any remark on that. How many were there MIPS classes used in the analysis? If they were many then p-value of 3.83E-03 (chaperones) may be well in the hell of insignificance. Why MIPS, FunSpec and year 2002? GOrilla allows to analyse a ranked single list of genes which appears preferable over setting an arbitrary threshold to select a target group (http://cbl-gorilla.cs.technion.ac.il/). Not only chaperones but also other categories have rather unimpressive p-values. A large data set often yields results at similar levels of uncorrected significance but with hardly any meaning.

Thank you for drawing our attention to the outdated choice for functional enrichment analysis. In the current manuscript we performed a new analysis using up-to-date GO Slim categories, retrieved from the Saccharomyces Genome Database. We write: “Based on the systematic genetic-genetic interaction screen, the list of genes showing negative interaction with the yEVenus overexpression (i.e. their deletions increased the fitness effect of overexpression) were retrieved and tested for Gene Onthology term enrichment with topGO (version 2.28) (Alexa et al., 2006) and org.Sc.sgd.db (version 3.3.0, (Marc Carlson, 2016)) packages in R programming environment (R Core Team, 2017). To focus on the important GO terms we restricted our search to the GOSlim categories maintained by the SGD project (Cherry et al., 2012). A GO category was termed as enriched significantly, if the genes annotated to a particular GO term were significantly overrepresented (p < 0.05, Fisher's exact test) in the given gene set using the complete list of screened genes as background.”

The new Table 1 also contains FDR correction for the multiplicity of testing. Please note that, based on the FDR corrected p-values, the category of protein folding remains enriched within the negative interacting genes.

B) Translation, transcription, metabolism of RNA and mitochondria show up as enriched in nearly every study of growth effects caused by deletions under nearly any conditions. I know that here the idea is to extract interaction out of joint effect of two factors. But, I would be careful.

We strongly believe that enrichment of the above cellular processes is not a pure coincidence, as environmental perturbation of these processes reassuringly affirmed the results of the genetic interaction screen. In addition, these results are consistent with previous studies.

What categories were enriched among deletions with positive epistatic effect? Have I missed these results?

Indeed, we did not put an emphasis on genes with positive epistatic effect, mainly for two reasons. First and foremost, the main focus of this manuscript was on the genes showing negative epistatic effect, as such genes and the corresponding cellular processes buffer protein production cost. Second, we found a lack of functional enrichment among the genes with positive epistatic effect. This finding is now included in the revised manuscript.

C) The following statement is meant to justify why the paper is focused on co-translational chaperones: "The list (of gene deletions interacting with GFP overproduction; rev add) includes members of the Gim/prefoldin complex (GIM4/GIM5/PFD1), two Hsp90 co-chaperones (CPR7, STI1), a type I HSP40 co-chaperone involved in regulation of HSP90 and HSP70 functions (YDJ1), and a member of the Hsp70-Hsp40 chaperone complex (SSE1). The majority of these proteins are parts of a chaperone network involved in co-translational folding of thousands of client proteins. These so-called CLIPS chaperones (Albanèse et al., 2006) associate the translation apparatus with nascent polypeptides emerging from the ribosome and help de novo folding of newly synthesized proteins. Inactivation of these genes sensitizes the proteome to protein misfolding and downstream aggregation (Albanèse et al., 2006; Willmund et al., 2013)."My objections:1) CLIPS is not a real, functionally defined, class of proteins. It is just an operational term covering genes involved in chaperoning/folding functions which are down-regulated in their expression at the time of stress (sometimes transiently) as opposed to HSP, which are up-regulated, and was used just in one (?) publication (Albanese, 2006). The CLIPS proteins are only indirectly linked to participation in chaperoning of nascent proteins: translation is halted (transiently) when stress comes, so CLIPS may be involved in chaperoning fresh polypeptides. (Besides, STI1 is in HSP, SSE1 and YDJ1 are in between, CPR7 and FPR4 work on proline twisting together with chaperones but are not chaperones themselves.)

In the current version we revisited the results and focus on Hsp70/90 chaperone network only.

2) In fact, none of the listed proteins is working co-translationally. Prefoldin sounds nice but it just assists selected new proteins on their way to TRiC after they are released from a ribosome. SSE1 is somewhat linked to nascent proteins, it is a nucleotide exchange factor for SSB (co-translational Hsp70). But it works also for SSA (stress-induced Hsp70). Besides, SSE1 releases, not binds, clients of SSB, possibly after they leave ribosomes (Willmund).3) The co-translational binding of SSB to growing polypeptides is probably facilitated by ZUO1 and SSZ1, it is a real pity these two most relevant genes are not among those spotted in this work (SSB1 and 2 had to be deleted both, one deletion does not have any effect).4) Although SSBs are believed (rather than truly confirmed) to chaperone co-translationally, their action is poorly recognized and probably not universal. Willmund says: "Thus, co-translationally acting Hsp70 meets the challenge of folding the eukaryotic proteome by stabilizing its longer, more slowly translated, and aggregation-prone nascent polypeptides". Authors of this manuscript say that their GFP is codon-unbiased, and it is just 230 aa long, so it does not appear a suitable client for SSBs anyway. Sure, overproduced GFP can misfold and aggregate more often than when rare but unfolding of such stuff would most probably involve SSAs and many other, non-co-translational chaperones. Where are they?

Thank you for the clear descriptive summary on chaperones. After revisiting the results of genetic interaction screen we now highlight the importance of the Hsp70 SSA complex regulation in the prevention of proteome misfolding in times of protein overexpression. We write: “The genetic interaction screen revealed that molecular chaperones are overrepresented in the list of genes that influence protein production costs. Most notably, the list includes several members of the Hsp40-70-110 complex (FES1, SSE1 and YDJ1), and an Hsp70-90 scaffold protein (STI1). These Hsp70-associated proteins are functionally highly related, and all play critical roles in the ATPase activation and the nucleotide exchange regulation of the Hsp70 class Ssa chaperones (Figure 5).” We also write: “It is worth noting that due to partial functional redundancy of SSA proteins (Hasin et al., 2014), the corresponding SSA genes did not emerge in the screen.” Accordingly, we propose that perturbation of the SSA chaperone complex – by inactivating its key regulators – renders yeast hypersensitive to the overexpression of a gratuitous protein.

[Editors' note: the author responses to the re-review follow.]

Essential revisions:1) Examine and discuss the complete set of results from the unbiased genome-wide approaches (the genetic interaction screen and the co-IP screen), not just the set of results that can be rationalized. In our opinion, these 2 approaches provide most of the novelty of this work, and so their complete results should be more prominent in the paper. For example, the nature of positive genetic interactions with YFP should also be examined for completeness. Even if no significant enrichment could be detected among the genes involved in positive interactions, the authors should still report the observation. While the authors' focus on chaperones is understandable, as it constitutes an additional cost that can be easily rationalized as part of the protein's production history, I felt distinctly short-changed by the absence of further study (or even discussion!) of other genes that revealed negative epistatic interactions. This includes genes related to mitochondrial organization/translation, explicitly listed as enriched in Table 1, but also other genes found that do not fall into significantly enriched GOSlim categories and remain entirely unmentioned.

Thank you for your understanding the need to keep the paper focused on chaperones. We restrained ourselves from wild speculations, and attempted to explore the scenarios raised in the paper in detail.

As regards positive interactions, we could not resist to mention one interesting case. A specific repressor of the Ras-cAMP pathway (RPI1) showed strong positive genetic interaction with yEVenus overexpression. Although the underlying molecular mechanism remains to be studied, this pattern may have some biotechnological relevance.

As regards the role of mitochondrial genes, we added a new paragraph. We write:

“Moreover, genes with mitochondria related functions, including mitochondrial, translation (e.g. MRPS9, MRPL22), mitochondrial DNA replication and growth (e.g. MMM1), mitochondrial distribution and morphology (e.g. MDM38) are on the gene list identified by the SGA analysis (Supplementary file 1).”

In the Discussion section we add:

“We note that mutants with impaired mitochondria exhibit reduced respiratory growth, and therefore they have to rely on less efficient modes of ATP production. However, beyond ATP production, mitochondria are involved in the synthesis of certain amino acids as well (Ahn and Metallo, 2015; Zong et al., 2016). Therefore, future works should elucidate the exact molecular mechanisms underlying the elevated protein burden in cells deficient in mitochondrial functions.”

Two additional experiments:2) Express yEVenus from a different promoter. (Rationale: The explanation that the overexpression of YFP perturbs chaperone network by "sequestering" parts of it (as Figure 4 suggests) is appealing. However, the overexpression of YFP in the study is driven by the promoter of HSC82, a chaperone. Although the authors claim that it is relatively constitutively expressed, querying SPELL for expression profile shows that HSC82 is co-expressed with many other protein folding proteins, suggesting that it shares transcription factors with them. Adding multiple additional copies of the HSC82 on a multi-copy plasmid could therefore decrease expression of many chaperones simply by competition, making cells more sensitive to deletion of particular chaperones. Therefore, it is important to confirm at least the most important results of the paper by overexpressing YFP using a different promoter that is unrelated to chaperones.)

This is an important objection. As requested, we overexpressed yEVenus under the control of native promoters of genes with non-chaperone functions, including pGPP1, pTAL1, pPDC1, and pTDH3. These promoters drive the expression of cytosolic proteins with similar cellular abundances as Hsp90p (Wang et al., 2012), see also Figure 3—figure supplement 1). Two results confirm the conclusions of the paper:

First, gene overexpression cost significantly increased with rising temperature in each promoter-yEVenus combination (Figure 3—figure supplement 1).

Second, the negative genetic interaction between yEVenus overexpression and specific chaperone regulators remained regardless of the promoter employed (Supplementary file 2, Figure 3—figure supplement 1). The analysis was performed on strains with deficiency in STI1 and YDJ1 (specific regulators of the HSP70 complex), not least because the protein interaction analyses focused on the corresponding proteins.

3) Reciprocal yEVenus-Sti1 Co-IP. (Rationale: Given the major novel finding of the manuscript relies on the chaperone network having a critical role in shaping protein production costs, it would be nice to show in the main text a validation of the physical interaction between yEVenus and Sti1 through reciprocal co-immunoprecipitation analysis. Further, their Co-IP mass spectrometry data identifies several other cellular proteins that physically interact with yEVenus and do not have a role in chaperone function, and this analysis does not identify any other chaperone related genes, which based on their conclusions in surprising. The authors make no comment of these inconsistent observations in the text.

The table we presented in the first submission contained a handful of proteins that met very stringent criteria we used to filter the results of the co-IP assay. It was not our intention to claim that yEVenus interacts only and exclusively with these proteins. Rather it allows us to focus on what we consider the most interesting hit (Sti1p). Please note also that we identified several other chaperones that putatively bind to yEVenus when overexpressed (Supplementary file 3). One slight modification in the revised manuscript is that we deleted discussing interactions under proteotoxic stress, not least because only one chaperone was detected to bind to yEVenus in a condition specific manner.

Additionally, we performed a reciprocal co-immunoprecipitation analysis, and found evidence that yEVenus perturbs binding between Sti1p and their native interactions partners (Figure 4).

We write:

“The analysis focused on 18 proteins, all of which have been described to physically interact with Sti1p in prior studies (Cherry et al., 2012). […] Collectively, these data suggest that protein burden promotes a partial disassociation of these proteins from Sti1p, putatively leading to partial disassociation of the Hsp70-Hsp90 chaperone complex.”

Additional points raised by one or more reviewers that should be addressed:4) The authors identify 184 gene deletions that show confident negative genetic interactions with overexpression of YFP. The authors report that this list of genes is enriched for ribosome biogenesis, transcriptional control, amino acid metabolism, mitochondria-related processes and protein folding (Table 1). However, none of the GO terms listed in Table 1 seem to be associated with ribosome biogenesis or amino acid metabolism. Could the authors clarify where that conclusion originated?

This was indeed an unfortunate sentence. We modified it as follows:

“Protein synthesis is frequently limited by the availability of free ribosomes (Vind et al., 1993). […] In agreement with expectation, partial inhibition of translation elongation by cycloheximide treatment elevated protein burden (Figure 2).”

We also write:

“Another source of protein burden may arise due to wastes of cellular resources, including ATP and amino acids needed for protein synthesis. Indeed, inactivation of amino acid metabolism genes (AAT2, BAT2, CYS3, PRS3, LEU3) influenced protein burden (Supplementary file 1), suggesting that protein burden depends on the availability of amino acids in the environment. It was indeed so: depletion of amino acids in the growth medium increased the fitness cost (Figure 2).”

Also, the enrichment for mitochondria-related processes is never discussed in the manuscript.

In the revised version, we discuss it as requested. For more details, see response to the first comment.

5) One of the main motivations for the study was provided by the genome-wide genetic interaction screen. The goal of this screen was to uncover gene deletions that mitigate or exacerbate the fitness cost of unneeded protein overproduction. A key assumption is that the protein load (i.e., the number of YFP molecules per cell) is approximately the same in all ~5000 deletion mutants and thus any potential fitness defects are due to the inability of the deletion mutant to cope with the load. The justification for this assumption is provided in Figure 1 where the authors plot the correlation between the genetic interaction score (i.e., the synergistic fitness gain or defect resulting from combining a deletion with YFP overexpression) as a function of "biomass-normalized fluorescence level" (i.e., YFP abundance). Given the lack of correlation, the authors conclude that "genetic interactions reflect a change in the fitness cost, but not in the extent of protein overexpression".I'm not sure that Figure 1 addresses this question completely because it is not clear what "biomass-normalized fluorescence level" actually represents. Biomass normalization is done by dividing the total fluorescence of a yeast culture by its OD. However, OD depends both on cell number and cell size/volume, which vary across deletion mutants and may affect the total culture fluorescence in a complex way. The authors should therefore discuss this relationship in more detail and make sure that normalized fluorescence does indeed represent the extent of protein overexpression. This is especially important considered that, according to Figure 1, normalized fluorescence varies extensively across mutants. What is the most likely cause for that variation? For example, a number of mutants are known to harbor more copies of high-copy plasmids than others (e.g., bik1, bim1) – do those mutants tend to show lower normalized fluorescence?

Optical density normalized fluorescence is probably the most feasible way to estimate the level of protein production in a systematic manner (i.e. on the complete haploid yeast deletion collection). However, as the reviewer indicated, cell size and cell volume may affect optical density and therefore the cell number of a culture. Therefore, we performed the same analysis by excluding genotypes with extreme cell size (top 5% largest and smallest cell size according to Jorgensen et al., 2002): the correlation between genetic interaction score and biomass normalized fluorescence remains very weak (Pearson’s correlation test, r = 0.035, p = 0.02, see Figure 1—figure supplement 1). Reassuringly, a partial correlation analysis also shows that the correlation between genetic interaction score and biomass normalized fluorescence remains weak (adjusted R^2^ = 0.0031) after control for cell size as a continuous variable (Ohya et al., 2005).

We modified the text as follows:

“Biomass-normalized fluorescence level had no major impact on the distribution of genetic interaction scores (Figure 1). This pattern was not due to any major deviation from wild type cell size (Figure 1—figure supplement 1) This indicates that genetic interactions reflect a change in the fitness cost, but not in the extent of protein overexpression.”

6) The identification of Sti1 as a protein that not only influences YFP production but also physically binds YFP is interesting. However, it is important to verify how often Sti1 co-purifies with other proteins, i.e. the number of physical interactions currently reported for Sti1 in the databases.

To investigate this issue systematically, we performed a reciprocal co-IP assay. We focused on 18 physical interaction partners of Sti1p described in the Saccharomyces Genome Database (SGD, (Cherry et al., 2012)), and confirmed 9 (50%) of them in our assay. In a nutshell, the analysis revealed that YFP (or more precisely yEVenus) overexpression perturbs native interactions of Sti1p (See Figure 4).